# Immune Cell Migration to Cancer

**DOI:** 10.3390/cells13100844

**Published:** 2024-05-16

**Authors:** Allison T. Ryan, Minsoo Kim, Kihong Lim

**Affiliations:** 1Department of Microbiology and Immunology, University of Rochester, Rochester, NY 14642, USA; allison_ryan@urmc.rochester.edu (A.T.R.); minsoo_kim@urmc.rochester.edu (M.K.); 2David H. Smith Center for Vaccine Biology and Immunology, University of Rochester, Rochester, NY 14642, USA

**Keywords:** immune cell, migration, trafficking, leukocyte, cancer, tumor, tumor microenvironment, T cell, myeloid-derived suppressor cell, MDSC, chemokine, chemotaxis

## Abstract

Immune cell migration is required for the development of an effective and robust immune response. This elegant process is regulated by both cellular and environmental factors, with variables such as immune cell state, anatomical location, and disease state that govern differences in migration patterns. In all cases, a major factor is the expression of cell surface receptors and their cognate ligands. Rapid adaptation to environmental conditions partly depends on intrinsic cellular immune factors that affect a cell’s ability to adjust to new environment. In this review, we discuss both myeloid and lymphoid cells and outline key determinants that govern immune cell migration, including molecules required for immune cell adhesion, modes of migration, chemotaxis, and specific chemokine signaling. Furthermore, we summarize tumor-specific elements that contribute to immune cell trafficking to cancer, while also exploring microenvironment factors that can alter these cellular dynamics within the tumor in both a pro and antitumor fashion. Specifically, we highlight the importance of the secretome in these later aspects. This review considers a myriad of factors that impact immune cell trajectory in cancer. We aim to highlight the immunotherapeutic targets that can be harnessed to achieve controlled immune trafficking to and within tumors.

The development of a productive immune response depends on the finely regulated and coordinated activity of immune cells within the body. The innate immune system is activated first to broadly neutralize pathogens, followed by the adaptive immune system, which imparts an antigen-specific response. These defense systems synchronize to clear pathogens and restore homeostasis. During both homeostasis and infection, the coordinated activity of immune cells is largely dependent on sensing environmental cues, intercellular communication, delivery of effector cytokines, and cell surface receptor expression. Of equal importance is how immune cells can integrate dynamic signals in various environmental conditions. These factors converge on the integral cellular activity that results in the functional localization of immune cells, otherwise known as migration.

Not only must a cell be properly activated to respond to an immune insult, but it must also orient to the right location at the right time. Motility, migration, and trafficking represent the complex biological activities that contribute to this essential aspect of a functional immune system. These immune cell dynamics become even more complex within the context of cancer and in the tumor microenvironment. While some intricate details of the mechanisms at play remain unknown, there is a great deal of knowledge available to aid in the progression of our understanding. In this review, we highlight current knowledge of innate and adaptive immune cell migration in the context of cancer, and, finally, translational applications derived from these recent observations.

## 1. Basic Concepts of Immune Cell Movement

### 1.1. Immune Cells Employ Varying Modes of Migration

To perform essential immune surveillance and facilitate cellular interactions, immune cells must use different types of motility and migration to navigate through dynamic environmental conditions, reach inflamed sites, and ensure proper proximity with their immune counterparts and their targets (Figure 1). First, the term motility here refers to a cell’s intrinsic ability to move, an otherwise energetically demanding phenomenon, which is partly dependent on metabolism [1,2]. Without additional cues, this represents stochastic, cell-autonomous migration events that enable cells to explore and sample their surroundings to respond quickly and effectively to threats. These movements require adhesion molecules which include integrins (discussed below). Second, immune cells also exhibit directional migration, guided by external signals that arise from tissues. Signal-guided migration includes chemo-, hapto-, and durotaxis, among others reviewed elsewhere [3,4,5]. The most common type of signal-guided migration is chemotaxis, which results from the sensing of chemical cues known as chemoattractants. Chemotactic movements occur when differences in chemoattractant concentration are sensed between a cell’s two opposite ends. The outcome is directional migration along a gradient to an area of interest, often inflammation or infection [3]. Alternatively, haptotaxis is the response to substrate-bound/immobilized chemical cues [3]. Cells also move in response to physical guidance, including toward stiffness, which is termed durotaxis [3]. Third, by a different classification of migration, cells can adopt mesenchymal, ameboid, or lobopodial migration [6]. Mesenchymal migration is a motility pattern dependent on adhesive interactions between the cell surface and substratum characterized by actin mesh-rich leading edge and protrusions [4]. Ameboid migration is a flexible and dynamic locomotion that enables cells to patrol complex three-dimensional surroundings by squeezing through small spaces within extracellular matrix. This migration pattern is driven by actomyosin network remodeling at the cell front and rear, triggered by chemoattractants binding to their G-protein coupled receptors (GPCRs) on the cell surface [7]. Lobopodial migration is considered a hybrid between amoeboid and mesenchymal, exhibiting tight adhesions with asymmetric protrusions at the leading edge called lobopodia [4,8]. Overall, cell migration patterns are determined by the porosity of ECM, the degree of confinement, the local concentration and distribution of ligands that a cell binds, and the intrinsic cell state (e.g., expression levels of adhesion molecules and secretion of proteases). Importantly, a cell can shift between these types of migration multiple times during one single journey to its destination. An extensive review of immune cell migration patterns can be found elsewhere [4].

### 1.2. Adhesion Molecules Are Required for Immune Cell Migration

#### 1.2.1. Integrin and Integrin Ligands

Integrins are heterodimeric transmembrane proteins comprised of α and β subunits. A total of 18 α and 8 β subunits are expressed in vertebrates, with canonical and non-canonical combinations expressed on the surface of leukocytes and inflamed endothelium. Cells anchor within tissues via integrin binding to extracellular matrix (ECM) and can physically interact with other cells via integrin binding to surface proteins of adjacent cells. Ligand binding to integrin also results in intracellular signaling cascades that influence cell fate and function [9]. Therefore, integrin expression and integrin-ligand interactions are fundamental determinants of host immune function via immune cell trafficking throughout the body, localization and retention of immune cells within tissues, and immune cell-target binding [10]. While all the integrin subtypes play important immunological roles, β2 integrins are particularly important for leukocytes in broad physiological contexts.

LFA-1 (αLβ2 or CD11a/CD18) plays critical roles in immunity, as it is required for adhesion, migration, and activation of immune cells [11,12]. While GPCR-mediated intracellular signals control LFA-1’s ability to bind its ligands, downstream signaling pathways initiated through LFA-1 confer highly conserved and dynamic responses that facilitate immune interactions. The importance of LFA-1 and its ligands for T cell polarization and migration are discussed in detail by Walling BL and Kim M [13]. LFA-1 has several relevant ligands. Intercellular adhesion molecule 1 (ICAM-1) is the canonical ligand to LFA-1 that regulates T cell migration, especially in terms of vascular and transendothelial migration, thus the binding of ICAM-1 to LFA-1 enables T cell tissue infiltration. Interaction of ICAM-1 and LFA-1 is also important for longer T cell-antigen presenting cell (APC) interactions and improved target cell killing [13]. Additional ligands for LFA-1 include ICAM family members such as ICAM-2 and ICAM-3, among others [12]. ICAM-2 is expressed on lymphocytes, monocytes, and endothelial cells, and is similarly important for endothelial-immune interactions. ICAM-2 also seems to play an important role in NK cell clearance of tumor cells. Additionally, ICAM-3 (also binds CD11d/CD18 or αDβ2) is expressed by leukocytes, but also can be found on endothelium and cancer cells. Literature regards ICAM-3 as a major component of adaptive immune activation, critical for naïve T cells to establish initial contact with APCs. These ICAMs are discussed in great detail by others [14,15].

MAC-1 (αMβ2, CD11b/CD18) is expressed by myeloid cells and is activated by chemokine signals and E-selectin-glycan interactions [16]. The expression of MAC-1 on innate immune cells is very important for immune surveillance and pathogen clearance [17]. Since myeloid cells often express LFA-1 and MAC-1 simultaneously and ICAM1/2 are ligands for both integrins, it was unclear to what extent each of these integrins contributed to myeloid cell migration. Phillipson M et al. clearly delineated LFA-1 and MAC-1 are non-redundant in neutrophil transendothelial migration, each with a unique contribution to specific steps of the transendothelial migration [18]. MAC-1, unlike LFA-1, has many potential ligands including ICAM-1, ICAM-2, ICAM-4, iC3b, fibronectin, fibrinogen, heparin, collagen, and more [12]. This promiscuity represents the diverse immunological functions that innate immune cells can facilitate through MAC-1. 

Another integrin subtype of major significance to leukocyte adhesion and migration includes VLA-4 (α4β1). Leukocytes can bind Vascular adhesion molecule 1 (VCAM-1), MAdCAM-1, fibrinogen, fibronectin and more via VLA-4, thus VLA-4 is involved in the migration and development of T cells, B cells, neutrophils, and monocytes. Like ICAMs, VCAM-1 has other binding partners, including αDβ2, which further highlights the diversity of integrin ligands that can contribute to immune localization and cell-cell interactions [12]. 

There are even more integrin subtypes of significance to leukocyte adhesion and migration which can be explored through the extensive existing body of literature [12]. Overall, integrin ligand density, and expression levels and affinity of integrins are key regulators of integrin-coordinated immune responses. These concepts have been cohesively reviewed by our group and others [7,12,13].

#### 1.2.2. Selectin and Selectin Ligands

Another important component of immune migration includes selectins. Selectins are single-chain transmembrane glycoproteins that resemble C-type lectins, found on the surface of immune cells and endothelial cells, especially within the vascular network. They function by mediating inflammatory responses via the ligation of glycoproteins present within the environment [19]. L-selectin (Leukocyte-selectin, CD62L) is required for naïve T and B cell entry into lymph nodes via the high endothelial venules (HEV) [20]. These proteins are essential for T cell homing and enable effector function in the periphery. Importantly, selectin expression on CD4^+^ T cells is modulated by cytokines, which alters T helper (Th) subset differentiation and migration patterns, especially in the skin and gut [21]. Expression patterns of P-selectin (Platelet-selectin, CD62P) and E-selectin (Endothelium-selectin, CD62E) are also modulated by cytokine patterns and are important for immune cell migration, especially innate immune cells including neutrophils, monocytes, and dendritic cells (DCs). Selectin expression on platelets and endothelium contributes to tissue accessibility for immune cells [22,23]. Selectin ligands are glycosylated proteins present on the surface of cells. Selectins and their ligands were reviewed considerably in the context of cancer by Witz IP [24]. Overall, selectin expression is important for immune cell activation by permitting entry and egress from lymphatics and vasculature and contributing to antigen-specific activation of effector immune cells. 

In summary, we have outlined the major proteins that regulate immune cell migration. The importance of adhesion molecules and their ligands in inflammation, tissue repair, and immune surveillance cannot be understated. We continue to learn more about the immunologically relevant dynamics of this highly nuanced phenomenon every year. 

### 1.3. Migration Patterns of Specific Immune Cells

While adhesion proteins are important regulators of migration for all cell types including leukocytes, each subtype exhibits their own general patterns of migration, governed by specific signaling events. Importantly, neutrophils, monocytes, and DCs are critical innate responders. Neutrophils express integrins including LFA-1, MAC-1, VLA-4 and GPCRs including chemokine receptors for their migration and trafficking. LFA-1 and MAC-1 are the most important integrins that govern neutrophil mobilization—from bone marrow (BM) egress, transendothelial migration, interstitial migration, tissue localization, to target killing. Please refer to excellent reviews by other groups for more detail [25,26,27]. The egress of neutrophils from BM to blood is regulated by the opposite action of CXCL12 secreted by BM stromal cells and neutrophil attractants in blood. Increased levels of CXCL chemokines in inflamed hosts drive more BM neutrophils to migrate out to blood compared to healthy hosts. Furthermore, G-CSF generated at the sites of inflammation also supports neutrophil mobilization [28]. They then rely on Leukotriene B_4_ (LTB_4_), complement, and CXCLs (e.g., CXCL1, CXCL2, CXCL8) to perform their effector functions in the periphery [28]. Certainly, many factors contribute to the regulation of neutrophil mobilization in the BM and periphery. This is true also in the context of cancer, and those mobilized neutrophils make significant impacts on cancer growth and development. We touch on this later. 

Monocytes express the same battery of integrins as neutrophils, LFA-1, MAC-1, and VLA-4. Detailed reviews discussing monocyte recruitment and migration are available elsewhere [29]. Monocytes rely largely on CCR2 signaling for BM egress, and then give rise to macrophages and dendritic cells in the periphery and are thus critical components of the innate immune system. Monocyte subtypes have different chemokine receptor expression patterns. Classical monocytes (also called inflammatory monocytes) express high levels of CCR2 and intermediate levels of CX3CR1 whereas nonclassical monocytes (also called patrolling monocytes) express no CCR2 and high levels of CX3CR1. Monocytes, like any other immune cells, infiltrate into peripheral tissues by sensing chemoattractants emanating from the tissue, later relying on cytokines and Toll-like receptor (TLR) ligands present in the tissue environment to orient themselves and exert their effector functions [30]. Their contributions to tumorigenesis, metastasis, and immunosuppression are highly significant, and we discuss examples of this below. 

DCs, represented within the immune system by heterogeneous subpopulations, are major governors of immunity due to their essential role in antigen presentation. DCs are dependent on integrins for their immune functions. Each subpopulation has different levels of integrin expression and employs a specific set of integrins for each of three critical processes: migration, tissue localization and retention, and effector function [31]. DCs rely on the expression of TLRs and other pattern recognition receptors to sense their environments. Once activated, DCs upregulate costimulatory molecules such as B7 proteins and migratory DCs move from peripheral tissue to lymph nodes to present antigens to adaptive immune cells. This population also pumps important cytokines into their environment to stimulate T cell expansion and differentiation. In addition to these three critical signals (antigen presentation, costimulatory activation, and cytokines), proper T cell positioning is also required for T cell activation. Chemokines orient DCs and T cells within lymph nodes and peripheral tissue, enabling antigen-specific T cells and licensed DCs to find each other. This guided interaction of DCs and CD8^+^ T cells can be mediated by chemokines from the environment (e.g., fibroblastic reticular cells) or auxiliary immune cells (e.g., CD4^+^ Th cells). Antigen-bearing DCs also produce chemokines to draw in CD8^+^ T cells directly, and to enhance DC-T cell contact [32,33]. Importantly, DC activation of T cells underlies a critical line of antitumor immune defense.

Each of these innate cell populations serves as a crucial intermediary to T cells, which are required for the development of the adaptive, antigen-specific response. We expand further on these innate cells in later sections. 

T cells are the main cytolytic effectors and regulators of our immune system, with important contributions that include antiviral and anticancer immunity. In addition to cytotoxic defense, T cells are responsible for the recruitment and coordination of other effector cells, promotion of B cell responses, and establishment of memory populations. The dominant integrins that T cells express for transendothelial migration and effector functions are LFA-1 and VLA-4. T cells also express a variety of other integrins in subtype-specific, tissue-specific, activation-specific, and differentiation-specific manners to attain tissue localization, retention, migration, and effector functions [34,35,36,37]. For T cells, these extensive behaviors take hold after antigen presentation occurs. As T cells move through lymphoid tissue in search of antigen, they are activated and primed by APCs. At this point, T cells can reenter circulation, and begin the migration process to their target tissue. Once their final location has been reached, T cells trigger the recruitment of subsequent waves of effector cells and work to eliminate pathogens or tumor cells. The events relating to naïve T cell search for antigen within the secondary lymphoid organs and subsequent egress is largely dependent on CCR7 and S1PR1 sensing their ligands CCL19/21 and S1P, respectively [38,39]. Infiltration into inflamed tissues is reliant on chemoattraction of T cells that sense inflammatory chemokines such as CCL3, CXCL9, and CXCL10, to name a few [7,40]. For T cells to carry out their diverse functions, they must be capable of tissue-specific migration that is coordinated by antigen encounter and activation. 

Antibody-secreting effector B cells are responsible for imparting a neutralizing humoral response [41]. B cells also play key roles in lymph node remodeling during an immune response, particularly through lymphangiogenesis and vascularization, which support the recruitment of T cells and antigen-presenting cells (APCs) [42,43,44,45]. Importantly, antibody-independent effector B cells provide critical support to DC and T cell interactions within the lymph node, by presenting antigen, promoting effector T cell and follicular helper (Tfh) T cell responses, and facilitating germinal center development [46,47,48,49,50,51]. Each of these phenomena is dependent on B cell migration, trafficking, and motility. These important immune functions are governed by the expression of CXCR5, TLRs, complement receptors, and CD40L. A narrative review that covers the role of B cells in CD8^+^ T cell regulation was recently published that describes the importance of B cell activation for context-specific CD8^+^ T cell antigen presentation, and memory formation. Additionally, they cover the role of B cells in promoting short-lived effector cells that are crucial for a productive immune response, and ultimately memory [52]. B cells make critical contributions to multiple aspects of a robust immune response which should not be overlooked. 

## 2. Adaptive Immune Cell Migration to Cancer 

### 2.1. T Cell Migration to Cancer

T cells are the main effectors responsible for recognizing and eliminating transformed cells. Even so, there are cancerous cells that evade or overcome immune recognition, leading to the manifestation of cancer and solid tumors. As those tumors grow and evolve, tumor-associated antigens (TAAs) and damage associated molecular patterns (DAMPs) are released into circulation. These molecular signals are recognized by innate immune cells, and TAAs are taken up and processed by APCs that prime T cells. This effectively initiates the arduous migration of primed effector T cells to the tumor site. The process of T cell homing requires dynamic expression of integrins, adhesion molecules, chemokine, and cytokine receptors, also influenced by lipid receptors, oxygen sensors, and interactions with other cells and their environment. Overall, the sequence is complex, and we are still working to understand how these, and other factors come together to achieve tumor-specific homing. Still, decades of research have advanced our understanding of some essential mechanisms. Here, we outline current knowledge of these mechanisms, speculating on alternative interpretations. Later, we will discuss recent clinical and translational therapeutic strides derived from these observations. In effect, our knowledge of the immune system will continue to promote new therapeutic applications to alleviate cancer and many other immunologically related diseases, including autoimmune disease (Figure 2).

#### 2.1.1. T Cell Migration to Cancer in the Perspective of Integrins and Cell Adhesion Molecules

As in homeostasis, integrins and other cell adhesion molecules, along with their diverse ligands, play an indispensable role in cancer surveillance and clearance. A primary example is LFA-1 and ICAM-1, which directly enables the infiltration of CD8^+^ T cells to CT26 colon cancer and B16 melanoma following increases in local IL-6 and soluble IL-6R [53]. Many others also report that inflammatory cytokine expression remodels integrin/ligand composition in tumors to promote infiltration of tumor-reactive T cells [54]. In a CD20/EGFR CAR T cell model, ICAM-1 was shown to increase in an IFNγ dependent manner and permitted tumor islet infiltration by T cells [55]. A small molecular activator of LFA-1 and VLA-4 facilitated localization of tumor-specific T cells to the tumor, improving antitumor response [56]. In effect, multiple inflammatory cytokines promoted the T cell response via modulation of ICAM-1. However, our understanding of the complex nature of inflammatory cytokines within the tumor microenvironment (TME) remains perplexing, especially for pleiotropic nature of cytokines such as IL-12 and IFNγ [57]. Polarizing cytokines support the differentiation and maturation of immune cells, which is associated with alterations in integrin and chemokine receptor expression [58,59]. Due to the interrelatedness of cytokines and cell surface protein expression, it is challenging to fully isolate these factors when evaluating integrin and adhesion molecule, or ligand alterations within the tumor. This becomes even more complex when considering tumor type.

Vascular changes arise in the presence of some inflammatory cytokines and their receptors within the microenvironment. Interestingly, VCAM-1 can be exploited for immune evasion in cancer such as renal cell carcinoma, mainly by inhibiting CD8^+^ T cell infiltration [60]. This phenotype is completely abolished upon mutation of VCAM-1 binding sites. Furthermore, VCAM-1 is pro-metastatic in breast cancer and pediatric osteosarcoma [61,62]. These observations suggest that VCAM-1 largely promotes tumor survival. Nevertheless, VCAM-1 is important for immune cell trafficking, and the mechanism responsible for promoting tumor survival remains unknown. 

Other LFA-1 ligands exist, including ICAM-2. Together with CXCL17, ICAM-2 promotes pancreatic tumor infiltration of dendritic cells, which increases specific lysis of tumor cells by cytotoxic T cells [63]. In gastric cancer, ICAM-2 expression limits disease progression, with downregulation associated with metastasis, angiogenesis, and advanced disease [64]. Additionally, in a murine model of breast cancer, ICAM-2 upregulation in dendritic cell vaccine therapy correlated with improved tumor control, which interestingly exhibited lower ICAM-1 expression [65]. These recent discoveries of ICAM-2 indicate that it may be a good prognostic marker for cancer progression and immunotherapy response.

Selectins are critical mediators of physical interactions between immune cells and endothelium during extravasation and intravasation events. As these interactions allow immune cells to enter new territory during homeostasis, they play a similar role in cancer, too. Furthermore, while L-selectin partially determines T cell recruitment to lymphoid tissue and tumor sites [66], it is known that L-selectin, E-selectin, and P-selectin also govern T cell infiltration to non-lymphoid tissues, recruit monocytes to metastatic tumors, and promote cell-cell interactions [21,67,68]. While selectins’ role in extravasation and metastasis has been described [69], their contribution to immune regulation via myeloid-derived suppressor cells (MDSCs) is particularly interesting. MDSCs represent a distinct population of cells that can dampen the endogenous immune response, ultimately supporting tumor survival and growth. A compelling series of studies from Sharon Evans’ group revealed that circulating MDSCs can enzymatically reduce L-selectin expression on naïve T and B cells outside the tumor [70,71]. In conjunction, they showed that dysregulated selectin expression results in lower antigen-specific effector cell response and an increased magnitude of MDSC immunosuppression [71]. Parker KH et al. described a similar phenomenon of MDSC suppression by L-selectin downregulation, this time as a function of HMGB1, which promoted MDSC differentiation and subsequent macrophage interaction, limiting antigen presentation to T cells [72]. This downregulation of L-selectin on circulating T cells hampered blood-lymph node trafficking events, contributing further to the suppressive effects. Other researchers have shown that, within the TME, L-selectin expression can improve central memory T cell infiltration independent of lymph node homing by way of increased activation, resulting in better tumor control [73]. These studies highlight the duality of selectins in the promotion or elimination of tumors. Still, enhanced expression of selectins within the TME is associated with MSDC activity, cancer progression, and metastasis [24,71]. Taken together, we see that cancer can exploit the expression of selectins to evade and alter the anticancer immune response not only within the TME but systemically by restricting T cell access to lymph nodes [70,72]. Several reviews discuss the importance of adhesion molecules in regulating cancer cells themselves, especially in the context of invasion and metastasis [74,75]. Finally, recent studies have introduced a novel concept of cis-signaling in T cells, with evidence that a receptor may bind a ligand expressed on the surface of the same cell to induce signaling. The idea is that a cell can be the source of both the ligand and receptor involved in one signaling event (e.g., B7:CD28 or 4-1BB:4-1BBL cis-interaction on T cells). This phenomenon may promote T cell effector function within the tumor, especially in the context of exhaustion and checkpoint-related molecules [76,77]. It is important to note that challenges present when trying to distinguish effects that arise as a result of cis- or trans-signaling in some contexts. We are excited to see how this field continues to unfold.

#### 2.1.2. T Cell Migration to Cancer in the Perspective of Chemokines

The elegant trafficking of immune cells is orchestrated by chemokines, secreted proteins regarded as master regulators of cell movement [40]. Chemokine sensing is dependent on the expression of cognate chemokine receptors on the cell surface. When these receptors are bound, downstream signaling occurs through localized generation of phosphoinositides or the influx of calcium ions, resulting in the activation of small GTPases [78,79]. These GTPases include molecules like Rac and Rho, which are positioned at the leading and/or trailing edge of the cell body. GTPases are capable of sensing small changes in chemokine gradients, translating this information to cytoskeletal dynamics and the microtubule organizing center (MTOC), thus enabling polarization of the immune cell body and directional migration [78,79]. As such, chemotaxis and chemotactic gradients enable T cell migration to and within various tissues (Figure 3).

##### CCL19, CCL21

CCR7, with its ligands CCL19/CCL21, is largely responsible for lymph node homing and homeostatic leukocyte migration, most notably influencing CCR7^+^ dendritic cells and T cells [38]. In the TME, these chemokines can have both pro- and antitumor roles. Through activation of APC and T cell migration to and within tumors, CCL19 can stimulate effective immune surveillance [80]. CCL19-expressing fibroblasts contribute to an antitumor phenotype in lung carcinoma by promoting the accumulation and cytotoxicity of CD8^+^ T cells [81]. Conversely, also in lung cancer, CCR7/CCL19 can promote the upregulation of heparanase-1 in the TME, contributing to the migration and invasion of A549 tumor cells, and lymph node metastasis [82]. Heparanase-1 is responsible for the cleavage of heparan sulfate, a primary component of the extracellular matrix (ECM), the major network of proteins that provide tissue structure and support for cells to sit and migrate. Heparanase-1 is overexpressed by nearly all cancers and importantly remodels the ECM, liberating cytokines and growth factors bound to heparan sulfate. These factors promote tumor growth through angiogenesis and metastasis [83]. The link between CCL19 and heparanase-1 was revealed by Zhang Q et al. 2013, who showed that CCL19 expression induces specificity protein 1 (Sp1) which binds the promoter region of heparanase-1 [82]. 

Separately, many cancer cells express CCR7, linking the lymph node homing effects with cancer cell metastasis to lymphoid tissue. For this reason, CCL19 expression has been associated with the upregulation of adhesion molecules related to migration, the epithelial-to-mesenchymal transition (EMT), and metastasis in breast cancer, cervical cancer, pancreatic ductal adenocarcinoma, and ovarian cancer, among others [84,85,86,87]. The current consensus is that CCL19 may be a biomarker for metastasis and is associated with tumor promotion, however, the impact of CCR7 signaling in the TME is conflicting because it is expressed on tumor cells and immune cells [80,88]. 

A recent study using analysis of the cancer genome atlas (TCGA) and in vitro transwell migration assays suggested CCL19/CCR7 directly drives T regulatory cell (Treg) migration to gastric cancer [89], providing a role for the signaling axis in immunosuppression. On the other hand, a study using induced SRC-3 (steroid receptor coactivator-3) Treg cells in a breast cancer model demonstrated that cytotoxic T cells effectively infiltrated into tumors in a CCL19/21-CCR7 dependent manner [90]. An additional report shows that tumors have an immune evasion tactic, where tumor cell secretion of CCL21 results in remodeling of the TME that efficiently sequesters effector T cells and promotes a protumor phenotype [91]. This protumor phenotype is especially interesting considering the importance of CCR7 signaling in homeostatic lymph node homing. Certainly, the effect of CCL19/21 and CCR7 in tumors is context-dependent, and the mechanisms will continue to be clarified over time.

#### CXCL9, CXCL10, CXCL11

CXCR3 is one of the most important chemokine receptors for T cell migration. It is expressed by activated Th1 CD4^+^ T cells and cytotoxic CD8^+^ T effector cells, plus other cells including natural killer (NK) cells [92]. CXCR3 ligands, CXCL9/10/11, play important roles in the TME, that can both support and inhibit tumor growth [93]. Interactions between CXCR3^+^ T cells and APCs contribute to increases in the ability of effector T cells to infiltrate inflamed tissue and promotes both antigen-specific and non-antigen-specific T cell activation [94,95]. Within tumors, CXCR3 expression by immune cells is largely antitumorigenic, promoting effector T cell localization [96]. CXCL9 in combination with IL-12 in the TME supports cytotoxic T cell effector function and contributes to antitumor immunity [97]. CXCL9 is also associated with T cell infiltration of cutaneous tumors, resulting in tumor suppression [98]. Interestingly, H3K27 methylation imparted by the polycomb repressive complex 2 (PRC2) can repress CXCL9/10 expression, disrupting effector T cell trafficking in colon and ovarian cancer [99,100]. Of note, PRC2 is overexpressed in multiple types of cancer and current research focuses on elucidating how PRC2 and other epigenetic regulators of tumor immunity contribute to immunosuppression and cancer progression. 

A recent elegant spatial transcriptomic analysis of human lung cancer revealed spatially organized ‘immunity hubs’ where stem-like T cells reside with myeloid cells through CXCL10/11–CXCR3 axis [101]. This study and previous work implicate the CXCR3 axis in activation of effector T cells and the recruitment of additional effector T cells. Another recently published study using imaging-based deep learning showed that CXCL9-mediated intratumoral clustering of cDC1s and CD8^+^ T cells is protective against cancer via increased antigen presentation and activation [102]. 

In contrast, opposing reports suggest that CXCR3 ligands can promote tumorigenesis. For example, CXCL10 induced by IFNγ contributes to invasion and metastasis in human colorectal carcinoma by induction of matrix metalloproteinase 9 (MMP9) [103]. CXCL11 or CXCL12 can alternatively bind CXCR7 which promotes tumor development [104]. This is particularly interesting because CXCR7, unlike most other chemokine receptors, does not trigger intracellular calcium mobilization or cell migration. At the time of this study, CXCR7 was a relatively novel receptor for CXCL11 and CXCL12, but was later revealed to be a scavenger receptor, or atypical chemokine receptor (ACKR). As such, the mechanism of tumor promotion is perhaps related to the sequestration of these important chemokines (discussed below) [105]. Additionally, Treg cells can be recruited via CXCR3 and its associated ligands, which results in dampened effector T cell responses within the TME, fostering tumor growth and survival [106]. These studies emphasize the duality of CXCR3 axis signaling within the TME and hint at the difficulties that may arise in designing effective, targeted therapeutics. While we continue to expand our knowledge regarding the role of CXCR3 and its chemokines on tumors, we recognize that they can be protumor in certain contexts. 

##### CXCL12

CXCL12, which is also known as stromal cell-derived factor 1 (SDF-1), recruits cells that express the cognate receptor CXCR4, including T cells among others. The CXCL12/CXCR4 axis plays a role in T cell infiltration and intratumoral trafficking in many cancer types. In a model of murine B16 melanoma, CXCL12-rich stroma on tumors sequesters T cells and prevents them from reaching their targets, limiting their effector capacity [107]. A study found that increased expression of CXCL12 by pancreatic cancer cells limited the infiltration of cytotoxic T cells, thus promoting the protumor response [108], with another group suggesting some cancer cells in pancreatic cancer-bearing mice evade T-mediated attack by reducing CXCL12 expression [109]. Another recent study showed that hepatocellular carcinoma (HCC) upregulates CXCL12 to recruit Treg cells and tumor-associated macrophages (TAMs), promoting tumor progression and metastasis [110]. An interesting mechanism proposed recently suggests CXCL12 provides tumor control via T cell retention in tumors. CXCR4^+^ CD8^+^ T cells exit tumors through CXCL12^+^ tumor-associated lymphatic vessels. T cells are retained in the tumors by downregulation of CXCR4 upon encounter of intratumoral antigens, which ultimately promotes antitumor immunity and tumor control [111]. Tumors are proficient in exploiting CXCL12 to restrict T cell localization and effector function, and there is significant therapeutic potential here. These studies indicate that targeting the CXCL12/CXCR4 axis may be a viable therapeutic strategy to improve the antitumor immune response in a spectrum of cancer types, both on its own and in combination with immune checkpoint blockade. We discuss this more in the therapeutic targeting chapter, considering the extensive roles of CXCL12/CXCR4 in immune cell trafficking in and out of BM, and in non-immune cell migration. Overall, we highlight the complex nature of CXCL12 in the TME and its contextual importance.

##### CXCL13

CXCL13, like the chemokines discussed above, exhibits dynamic functionality within the TME to promote tumor growth and progression, or the antitumor response. Hussain et al. highlight the significance of coexpression patterns of CXCL13/CXCR5 by immune cells, cancer, and cancer-associated cells within the TME; along with other soluble factors, they emphasize their contextual importance to tumor growth or control [112]. Classically, CXCL13 coordinates B cells [113], which in combination with CXCL13 producing CD4^+^ T cells and other tumor-reactive T cells results in the formation of tertiary lymphoid structures (TLS) [114,115]. These TLSs promote the antitumor response in combination with checkpoint blockade [116], and tumor metastasis via the recruitment of IL-10^+^ regulatory B cells [117]. 

##### Other Chemokines/Receptors

As suggested above, chemokine decoy receptors or atypical chemokine receptors (ACKRs) scavenge free chemokines including CCL19/CCL21, CXCL11, CXCL12, and CXCL13 without triggering intracellular signaling events [118]. These receptors arise in local microenvironments, including tumors, and can contribute to the pro- and antitumor response. High expression of ACKRs that scavenge CCL19/21 in breast cancer was associated with improved patient survival and reduced rates of metastasis [119]. In consideration of factors highlighted in the previous sections, this may be due to the reduced effect on CCR7^+^ cancer cells as free ligands are sequestered. Again, in breast cancer, ACKRs that scavenge CXCL12 limited tumor growth and metastasis in CXCR4^+^ breast cancer cells possibly due to reduced neovascularization or disruption of CXCL12-guided egress through tumor-associated lymphatic vessels [120]. These receptors are not just scavengers, contributing also to the generation and maintenance of chemokine gradients within microenvironments, which has varying effects on cell migration patterns. Overall, the mechanisms of decoy receptors may be more complex than their initial portrayal. Research is active to support the understanding of the biology of ACKRs, especially how and when they arise; Samus M and Rot A have extensively reviewed ACKRs in cancer as of 2024 [118], but ongoing efforts will reveal novel mechanisms at work.

CCL5 is a chemokine which binds its receptor CCR5 with high affinity, recruiting various immune cells in response to infection and CD4^+^ T cells to the TME [40,121,122]. Within the tumor, CCL5 is derived from intratumoral myeloid cells, T cells, and cancer cells [122]. Expression of CCR5 or CCL5 is broadly linked to poor prognosis in cancer, due in part to the promotion of MDSCs, plus genetic and metabolic alterations to antitumor immune cells and cancer cells, angiogenesis, and ECM remodeling [123]. CCL5 expression by gastric cancer is correlated with tumor progression; a possible mechanism behind this association is the disproportional accumulation of CD4^+^ and CD8^+^ T cells, plus selective apoptosis of CD8^+^ T cells in the tumor [124]. This selective apoptosis of CD8^+^ T cells may be due to direct cell-cell interactions with cancer cells. Similarly, CCL5 in colorectal cancer plays a critical role in immune escape through mechanisms of Treg cell recruitment and apoptosis of CD8^+^ T cells, possibly mediated by direct cytotoxicity of the Tregs against CD8^+^ T cells, promoting tumor growth [125]. *CCL5* knockout mice exhibit increased antitumor immunity associated with a deficit of Th2 CD4^+^ T cells [126]. More work is being done to understand the divergent effects of Th1 and Th2 CD4^+^ T cells in the tumor microenvironment. There are also reports contradicting CCL5’s protumor effects. CCL5 can mediate antitumor immunity by specifically recruiting antitumor CD8^+^ T cells to tumor sites, serving as a reminder of the highly contextual effects of chemokines in cancer [127,128]. 

CXCL16 is an atypical chemokine that exists in a soluble form (sCXCL16) or membrane-integrated form (TM-CXCL16); its cognate receptor, CXCR6, is expressed by T cells [129]. CXCL16/CXCR6 plays a role in T lymphocyte recruitment, retention, and survival within tumor stroma [130,131,132,133]. Interestingly, CXCR6 is especially important for recruitment and retention of memory T cells in the TME [134,135]. CXCL16/CXCR6’s role in cancer is also multifaceted, with conflicting results reported [136,137]. Allaoiu R et al. report that myeloid activation of CXCL16-secreting fibroblasts recruits additional myeloid cells, triggering immunosuppression [137].

### 2.2. Migration of NK Cells and B Cells to Cancer

NK cells are the first line of defense against viruses and cancer, and importantly, do not require antigen presentation, and can impart cytotoxic effector function to target cells via recognition of stress proteins including lectins or NKGD2 [138]. These MHC-independent effectors protect from metastasis in skin, breast, lung, and liver cancer, especially in conjunction with immunotherapeutic intervention, such as monoclonal antibodies or checkpoint blockade. Mechanistically, this increase in NK-tumor permissiveness may be due to increases in neoantigen circulating DAMPs in concert with CD8^+^ T cell activity [139,140,141,142,143,144]. Importantly, neoantigens can arise because of therapy or immune activation which boosts treatment response as the immune system is able to recognize new circulating antigen epitopes (e.g., epitope spreading). A study in murine breast cancer C3L5 cells that expressed CCL19 revealed tumor rejection in an NK cell-dependent manner [145]. Importantly, CCL19 was not demonstrated to activate NK cells but was crucial for chemoattraction of NK cells and DCs, and ultimately, long term immunity was conferred through CD4^+^ T cells. CXCR4 expression on NK cells facilitates entry into tumors in response to CXCL12 [144]. NK tumor infiltration is required for CCL5 production and recruitment of cDC1 cells that promote CD8^+^ T cell immunity [146]. Tumor infiltration of NK cells and its outcomes have been discussed in detail in other review articles [147,148]. NK cells are currently being explored as an autologous immunotherapeutic tool but their efficacy appears limited due to inefficient trafficking to solid tumors [144]. Further research into NK-specific chemokine signaling may improve this efficiency. Importantly, tumor cells often develop mechanisms to downregulate FasL, TRAIL, and Fc receptors, which allows cancer cells to evade NK immunosurveillance [149]. Perhaps in the future new therapeutics can mitigate this suppression, however, NK-specific immune evasion tactics will need to be considered in the development of any NK-based immunotherapies, including NK-CAR cells.

B cells perform multiple immunological functions, including antigen presentation and cytokine production, but their unique function within the immune system is to generate antibodies [150]. They impart neutralization via both soluble and insoluble immunoglobulins (Ig) [46,47]. These Igs are also capable of fixing complement, which is important during tumor cell death, as DAMPs derived from dead tumor cells may activate the complement cascade [151,152,153]. Accumulating evidence shows that B cells are capable of infiltrating tumors and proceed to form close interactions with T cells and myeloid cells [51,154,155]. Mueller et al. showed CD14^+^ monocytes and DCs support the growth, proliferation, and survival of healthy B cells, and can even induce plasmacyte differentiation—an important component of humoral immunity. In the case of diffuse large B cell lymphoma (DLBCL), this remains true, where myeloid interactions produce a complex chemokine and cytokine milieu including CCL5, IL-2, IL-12, and IL-10 which recruit additional cells to propagate malignancy, providing survival and differentiation signals for cancerous B cells. The authors suggest that blocking CCL5 may be a reasonable approach to curb the advancement of DLBCL. Separately, single-cell sequencing revealed that neoadjuvant chemotherapy, a type of cancer treatment that uses drugs, surgery, or radiation before the main treatment modality, induced distinct tumor-infiltrated B cell populations to arise because of complement C3 activation and complement-CR2 signaling [154]. These ICOSL^+^ B cells that elicit antitumor T cell immunity depend on complement cascades in both mice and humans, and this important effector B cell subset emerges after chemotherapy, as a direct result of increased C3b fragments in the bloodstream. Further, exhausted tumor-infiltrated CD103^+^ CD8^+^ T cells express CXCL13 upon TGFβ stimulation, which can directly recruit B cells. These B cells can form TLS as a means of reinvigorating the antitumor immune response [156]. CXCL13 promotes the formation of TLS that provide antitumor support through interactions between B cells and T follicular helper (Tfh) cells, and via priming and expansion of effector T cells within the tumor. This directly links T cell activation within the tumor to B cell recruitment. Further analysis suggests that patients with CXCL13^+^ CD103^+^ CD8^+^ T cells may be more responsive to immune checkpoint blockade. It is important to note that TLS formation is not entirely mechanistically understood. Evidence in non-small cell lung cancer suggests that tumor type and stage can influence the tumor-permissive B cell population [157], and that populations that are initially antitumor can shift toward protumor as cancer becomes more advanced. Still, these mechanisms are not fully understood. B regulatory cells have recently been characterized within tumors, but the state of the field is still developing, and their functions have not yet been clearly enumerated [158]. Overall, increasing evidence suggests that intratumoral B cell populations are heterogeneous, especially after conventional therapeutics, and these populations can modulate T cell effector recruitment and activity within the TME, to promote pro- or antitumor responses [154,159]. A deeper understanding of the mechanisms governing B cells within the tumor may reveal important biomarkers that will contribute to more effective immunotherapeutic interventions in a patient-specific manner. Single-cell sequencing and spatial transcriptomics will be invaluable tools to meet these goals.

### 2.3. Other Mediators That Regulate Adaptive Immune Cell Migration to Cancer

While chemokines are major regulators of cell migration, many other factors contribute to the overall complexity. Numerous examples in cancer, especially those treated with radiotherapy, show that tumors may become fibrotic, thereby increasing tissue stiffness, and directly impacting the number of tumor-immune infiltrates [160]. This is partly due to effects on cancer-associated fibroblasts (CAFs), and the metastatic nature of cancer cells [161,162,163]. These structural changes not only cause immune cells to adopt different modes of migration, but also alter tissue accessibility, further limiting cellular interactions and diffusion. Together, these factors disrupt the TME. 

Interestingly, circadian rhythm is recognized to affect leukocyte migration both during homeostasis and in cancer [164]. While these processes are still being worked out, evidence of circadian effects in tissue-specific leukocyte migration has been described by Scheiermann’s group [165,166]. This is an active area of investigation within the circadian biology field. Hypoxia, acidosis, and immunosuppressive cytokines within the TME can inhibit cell migration and function within the tumor [167]. This effectively dampens the antitumor response through multiple avenues including metabolism. Tumor stroma can upregulate fibroblasts and extracellular matrix proteins to create physical barriers that restrict T cell entry or residence within tumors [168], which can result in immunologically cold tumors that are known to have a poorer prognosis compared to their immune-rich counterparts. Many aspects of the tumor vasculature contribute to leukocyte migration and infiltration, which can be modulated to promote the antitumor response; these have been reviewed nicely by Lanitis E et al. [169]. Without therapeutic intervention, these vascular alterations often support tumor growth and immunosuppression, which we discuss briefly below.

## 3. Innate Immune Cell Migration to Cancer

The innate arm of the immune system protects hosts from bacterial, fungal, and viral infection, mediates tissue homeostasis, and induces and regulates adaptive immunity [31,170,171,172]. The innate immune system is comprised of multiple key members. Neutrophils are the frontline defense to eliminate pathogens such as bacteria and fungi using their incredible ability of phagocytosis and reactive oxygen production, and they are critical immune regulators in various disease settings [170,173,174]. Monocytes are a myeloid cell type with high levels of plasticity, replenishing tissue macrophages and dendritic cells in homeostasis and supporting tissue-resident myeloid cell populations in response to disease [172,175]. Tissue-resident macrophages are sentinels that rapidly respond to pathogens and foreign bodies, resolving those situations or leading the host to initiate a cascade of inflammatory responses [171]. Dendritic cells are professional antigen presenting cells that induce and regulate adaptive immune responses [31]. 

Innate immune cells are contextually protumoral or antitumoral, with the protumoral versions referred to as Myeloid-Derived Suppressor Cells (MDSCs). It is generally accepted that tumor infiltration of protumoral innate immune cells is required for a range of tumorigenic processes, including tumor initiation, growth, metastasis, and paradoxically, immune evasion [176]. The core contribution of innate immune cells to tumor development is through suppression of cytotoxic CD8^+^ T cells. Innate immune cells deplete essential metabolites and nutrients that CD8^+^ T cells depend on [177,178,179]. Tumor-infiltrated myeloid cells also suppress antitumoral T cells by inducing T cell ‘exhaustion’ through the upregulation of immune checkpoint molecules within the tumor [180], by enhancing inhibitory Treg responses [181] or by secreting immunomodulatory mediators [180,182]. Furthermore, tumor-infiltrated innate immune cells promote angiogenesis or neovascularization which is essential for tumor sustenance and growth [183,184]. Those innate immune cells provide multiple proangiogenic substances inducing and facilitating endothelial cell proliferation and behaviors. They are also major sources of matrix metalloproteases (MMPs), and the MMPs remodel extracellular matrices and activate the precursor form of proangiogenic factors such as vascular endothelial growth factor (VEGF) and TGFβ [185,186]. Protumoral innate immune cells play essential roles for tumor cells to spread throughout the body. These innate immune cells help tumor cells acquire mesenchymal features (epithelial-to-mesenchymal transition, EMT) [187], invade neighboring tissues by producing proteases [188,189,190,191], and metastasize [192,193,194,195]. In addition, tumor-infiltrated myeloid cells play a critical role in the clearance of dying cells in and around tumors to ensure that the tumor does not become overtly inflammatory, thus suppressing the antitumor inflammatory immune response [196]. 

Cancer therefore recruits protumoral innate immune cells or recruit and reprogram myeloid cells to support its growth, which is a critical factor underpinning tumorigenesis [197,198]. Tumor-associated chronic inflammation drives the innate immune cell accumulation in TME and successful tumors develop a string of mechanisms to recruit, retain, prolong, and manipulate innate myeloid cell state and effector function [199,200,201,202]. In many types of cancer, the overall prognosis is strongly tied to the amount and quality of neutrophils and macrophages in their microenvironment [203,204]. Therapeutically, the importance of these protumoral innate immune components provides targetable control points that can suppress tumorigenesis and ultimately cure patients of tumor burden [205,206,207,208]. In this part of the review article, we focus on mechanisms by which tumors recruit and retain those myeloid cells (Figure 4). 

### 3.1. TME-Derived Chemokines

Chemokines are essential protein molecules that guide both immune and non-immune cells to target sites during homeostatic and inflammatory conditions [40]. Tumor and tumor-associated cells produce and secrete chemokines to attract and retain immune cells in their microenvironment. 

#### 3.1.1. CCL2 

CCL2 is a chemokine that attracts CCR2-expressing immune cells (e.g., inflammatory or classical monocytes) and is highly relevant for cancer development and poor outcomes [209,210,211]. One of the seminal studies about MDSCs demonstrated CCR2 was involved in mobilization of MDSCs from BM to blood and from blood to tumors, contributing to the accumulation of CCR2^+^ MDSCs in 3LL and B16 tumors. Although the depletion of CCR2^+^ MDSCs did not lead to suppression of tumor growth or of T cells, this could be due to the aberrant excessive accumulation of neutrophils in the tumors [212]. On the other hand, Lesokhin AM et al. has shown CCR2^+^ M-MDSCs promote tumorigenesis by limiting activated CD8^+^ T cell infiltration [213]. In this study, B16 tumor-derived GM-CSF expanded CCR2^+^ M-MDSCs in tumors, which suppressed tumor infiltration of tumor antigen-specific CD8^+^ T cells. Conversely, MDSC depletion resulted in a greater accumulation of tumor-limiting CD8^+^ T cells in the tumor. However, CCR2 was found to be involved in MDSC egress from BM but not migration from blood to the tumor. These two studies using the same tumor model reported somewhat different results. The most noticeable difference was that Lesokhin’s group did not detect neutrophil accumulation in tumors upon CCR2^+^ cell depletion, suggesting that the sum or quantity of MDSCs may be more important than types of MDSCs in some tumor cases. 

Nesbit M et al. found a clear correlation among tumor production of CCL2, monocyte/macrophage accumulation, vascularization, and tumor growth [214]. In this study, all 30 melanoma cell lines tested expressed CCL2, whereas normal melanocytes did not, and CCL2 expression was required for the melanoma cells to form tumors in SCID mice. One possible tumorigenic mechanism of monocyte/macrophages recruited by CCL2 was tumor vascularization. Monocytes cocultured with CCL2-producing melanoma cells expressed TNF; monocyte-derived TNF induced tubule formation of HUVEC cells in vitro. One more interesting finding in this study was that if the melanoma cell line was rendered to express very high levels of CCL2 (more than 10 times CCL2 than average cell lines required to form tumors) those tumor cells failed to grow longer than 14 days, undergoing rapid necrosis with massive inflammatory cell infiltrates. 

CCL2 is a factor associated with tumor aggressiveness. Rogic A et al. have reported that CCL2 is the critical molecule aggravating inflammatory breast cancer (IBC) which, while rare, is an aggressive type of breast cancer [215]. In this study, A3250 human xenograft model recapitulating clinical and histopathological features of human IBC was established. A3250 IBC tumors induced prominent regional inflammation where CCR2^+^ CD11b^+^ F4/80^+^ and CCR2^+^ CD11b^+^ Ly6C^+^ myeloid cells, not Ly6G^+^ neutrophils, were dominant immune cell types. Importantly, those monocyte/macrophage cells often directly contacted tumor cells. The A3250 IBC cells and human IBC tumors highly expressed CCL2 compared to non-IBCs, and *CCL2* knockdown (KD) markedly reduced A3250 tumor growth and lung metastasis. *CCL2* KD A3250 tumor exhibited higher levels of necrosis and lower levels of tumor proliferation with a striking reduction of F4/80^+^ macrophage accumulation in the tumor. However, *CCL2* KD did not affect tumor cell growth in vitro, indicating that CCL2 was the key molecular mediator to modulate innate immunity in favor of the IBC tumor development and aggressiveness. Similarly, CCL2 increases aggressiveness of gliomas. Brain-resident macrophages, microglial cells, are frequently found in glioma or glioblastoma, sometimes constituting as much as 45% of the human glioblastoma cell suspensions. Importantly, CCL2 expression in glioma correlates with poor prognosis. Platten M et al. showed CNS-1 rat glioma cell line engineered to express CCL2 grew more aggressively and had more microglial cell infiltrates when engrafted into the basal ganglia of Lewis rats compared to the CCL2-negative parental cell line [216]. The suggested mechanism was that tumor-infiltrated microglia support glioma growth by promoting microvessel formation. A different research group reported an alternative mechanism behind CCL2 modulation of glioblastoma growth. Chang AL et al. showed CCL2 was produced by glioma cells and microglial cells in human glioblastoma tissues [211] and, in GL261 murine model of glioblastoma multiform, microglia and CD11b^+^ monocyte/macrophage were the major sources of CCL2. CCL2 was responsible for tumor recruitment of CCR2^+^ MDSCs and CCR4^+^ Tregs, suggesting the critical role of CCL2 in the generation of an immunosuppressive TME. Although this study did not address if the CCR2^+^ CD11b^+^ cells were MDCSs based on functional assays or definitive MDSC phenotypes (e.g., high level of Arginase 1 expression) and how CCL2 deficiency abrogated Treg accumulation in the tumor, others have reported that TME and MDSCs promote Treg infiltration into tumors by secreting T cell-attracting chemokines [217]. However, it seems that the same chemokines could recruit both antitumoral and protumoral T cells. Therefore, Treg accumulation might not be the result of selective recruitment, but instead, Tregs might be selectively enriched after the unselective recruitment of heterogeneous T cell subsets. TME often exhibits an immunosuppressive cytokine milieu rich in TGFβ and IL-10, which nurtures Treg cells but discourages inflammatory T cells. 

CCL2 can also be essential for tumor metastasis. In a study using a PyMT mammary tumor mouse model, inflammatory monocytes induced tumor cell metastasis and the molecular mediator was CCL2 [203]. Intravenously administered Met-1 cells (a PyMT-induced mouse mammary tumor cell line) required the recruitment of inflammatory monocytes to pre-metastatic sites to colonize the lung. These monocytes helped tumor cells extravasate into the lung, and monocyte-derived VEGFα was essential for the process. Importantly, tumor cell-derived and lung stroma-derived CCL2 was critical for the monocytes to mediate mammary tumor metastasis, validated by systemic blocking of CCL2 and by *CCL2* gene knockout in the tumor cells. In the same report, CCL2-mediated monocyte recruitment to the lung premetastatic niche was also critical for spontaneous metastasis of orthotopically injected MDA-MB-231 human breast cancer cells. Finally, CCL2 blocking significantly improved the survival of breast cancer-bearing mice. This elegant study revealed, for the first time, a string of CCL2-triggered metastatic processes, CCL2–monocyte–VEGFα–tumor extravasation–tumor metastasis. The study results suggest monocyte recruitment could be a therapeutic target to treat and prevent metastatic breast cancer. Similarly, therapeutic blocking of CCL2/CCR2 axis against hepatocellular carcinoma inhibited the recruitment of inflammatory monocytes and tumor-associated macrophages, resulting in the reversal of the immunosuppressive TME and inactivation of antitumoral CD8^+^ T cell responses [210]. 

#### 3.1.2. CXCL8, CXCL1, CXCL2, CXCL5 

CXCL8, also known as IL-8, is the primary chemokine to recruit CXCR2^+^ neutrophils in inflammatory situations in humans, playing a pivotal role in anti-microbial immunity [218]. CXCL1 and CXCL2 are also essential chemokines that drive neutrophil migration in humans and mice [40]. CXCR2 is the chemokine receptor for CXCL chemokines, including CXCL1, CXCL2, CXCL5, and CXCL8, and is predominantly expressed on neutrophils (CXCR2 immunobiology has been discussed in depth elsewhere [219]). While it is accepted that CXCR2 is a highly neutrophil-specific protein, there are reports that non-neutrophil cells including glomerular endothelium [220], sensory neurons [221], and neuroendocrine cells in human prostate cancer [222] also express CXCR2. Despite contradictory reports [223], high CXCL1, CXCL2, CXCL8, CXCR2 levels and accordingly high neutrophil levels correlate with cancer aggressiveness and poor patient survival in general [224,225]. Early Studies recognized that tumor cell lines produced CXCL8 or its homologous factors [226] and CXCL8 could promote the growth and development of various tumors, including melanoma, gastric cancer, lung cancer [227]. The study conducted by Bellocq A et al. is one of the earliest reports that found a chemokine-mediated link between neutrophil inflammation and cancer [228]. Patients with bronchioloalveolar carcinoma (BAC) also had neutrophil alveolitis. Patients’ bronchoalveolar fluid (BALF) and their tumor cells produced CXCL8, the concentration of which correlated with neutrophil numbers and neutrophil elastase activities in BALF. Finally, neutrophil alveolitis was strongly associated with a poor outcome of patients with BAC. It is now a consensus that tumor-associated neutrophils (TANs) or PMN-MDSCs have a significant impact on tumorigenesis and CXCL8 and CXCL1/2 are of the most outstanding chemokines promoting protumoral immune responses [229,230]. TANs or PMN-MDSCs in the TME have a multitude of protumoral effects which are key to cancer development. 

Neutrophils recruited to the sites of cancer initiation promote carcinogenesis through the generation of DNA damaging or DNA altering agents [231,232,233]. Sandhu JK et al. reported neutrophils recruited by CXCL8 could promote tumor development by increasing genotoxicity [231]. In a mouse fibrosarcoma model, CXCL8 induced neutrophil infiltration into the tumor, and the magnitude of neutrophil infiltration correlated with genotoxic ROS (reactive oxygen species) and RNS (reactive nitrogen species), as well as the mutational burden of tumor cells. The mutation frequency (MF) was determined by measuring gene inactivating mutation rate in hypoxanthine phosphoribosyltransferase (Hprt) locus. Hprt is widely used as a marker of mutation by genotoxicity because it is a non-essential gene in which mutational events can be quantified on the basis of resistance to 6-thioguanine. Genomic instability is an important feature of tumor progression, and this study is one of the first reports to show that tumor-infiltrating neutrophils could be the direct cause of tumor genomic instability, potentially contributing to the emergence of aggressive tumor phenotypes.

Numerous studies have supported the importance of CXCR2 ligands-CXCR2 axis for cancer growth and development using genetic ablation of mouse *CXCR2* or blockade of CXCLs or CXCR2 [234,235,236,237]. In one study, based on elevated levels of CXCL1 and CXCL2 in orthotopic and heterotopic Lewis lung cancer (LLC) models, WT and *CXCR2* KO mice were challenged with LLC tumors. Growth of primary tumor was attenuated in *CXCR2* KO mice and, to a greater extent, lung metastasis of LLC was decreased by *CXCR2* KO [234]. Another study found PMN-MDSCs were the dominant infiltrating immune cell type in a prostate adenocarcinoma model driven by loss of *Pten* and *Smad4* and pharmacologic inhibition of CXCR2 impeded tumor progression [236]. Separately, prostaglandins and their receptors underlie endometrial adenocarcinoma. The study by Wallace AE et al. revealed that prostaglandin receptor stimulation in endometrial adenocarcinoma and normal endometrial cells led to CXCL1 expression [235]. CXCL1 blocking antibody treatment led to a significant decrease in neutrophil infiltration into the tumors in an endometrial carcinoma xenograft mouse model. Senescence is a protective mechanism to combat tumor growth in the event that oncogenes are aberrantly activated or tumor suppressor genes are lost. Importantly, CD11b^+^ Gr1^+^ myeloid cells antagonized tumor cell senescence in a paracrine manner by interfering with the senescence-associated secretory phenotype. CXCR2 antagonism reduced tumor-infiltrating Gr1^+^ myeloid cells and greatly increased antitumor efficacy of senescence-inducing agents [237].

CXCR2 and its ligand CXCL8 are key factors for inflammation-driven tumorigenesis. Papilloma induced by topical treatment of susceptible mice with 7, 12-dimethylbenz[α]anthracene (DMBA) and 12-*O*-tetradecanoyl phorbol-13-acetate (TPA) contained substantial numbers of MPO^+^ CXCR2^+^ Gr1^+^ MDSCs, and *CXCR2* KO mice were profoundly resistant to the tumor induction [238]. In the same study, CXCR2^+^ MPO^+^ (Myeloperoxidase) cells infiltrated spontaneous intestinal tumors in *Apc^min/+^* and *AhCreER; Apc^fl/+^; Pten^fl/fl^* mice and CXCR1/2 blocking pepducin treatment suppressed tumor infiltration of MPO^+^ cells and adenoma formation concurrently. In a study with a transgenic mouse strain engineered to express human IL-8 under the control of its own regulatory elements, IL-8 expression was highly inducible by inflammation and also increased in colorectal tumors induced by administration of azoxymethane (AOM) injection and dextran sodium sulfate (DSS) [239]. The transgenic mice had greater numbers of CD11b^+^ Gr1^+^ innate immune cells and developed more tumors than WT mice. Another study also showed CXCR2-mediated MDSC infiltration was required for colitis-associated tumorigenesis [240]. Knockout of *CXCR2* attenuated chemically induced colitis and the following tumorigenesis with reduced numbers of neutrophils in tumor sites. The major contributor of the colitis-derived tumorigenesis was CXCR2^+^ MDSCs based on the findings that adoptive transfer of WT MDSCs restored tumor progression. MDSC suppression of T cells was proposed to be the major mechanism in this mouse tumor model.

Literature has frequently reported the importance of CXCL8 for tumor angiogenesis [241,242,243,244]. Because the CXCL8/CXCL1/CXCL2–CXCR2 axis regulates almost exclusively neutrophil mobilization, the proangiogenic effects from these chemokines are likely to originate from neutrophils. Nozawa H et al. used RIP1-Tag2 transgenic mice that spontaneously develop pancreatic neuroendocrine tumors and demonstrated tumor-infiltrated neutrophils were the major source of MMP9 [184]. Further, ablation of neutrophils could slow tumor development and angiogenesis in the same manner as MMP9 inhibition or knockout could. Thus, they proposed increases in VEGF bioavailability as a result of neutrophil-derived MMP9 could be the mechanism underlying the angiogenic influence of neutrophils. Additional studies showed tumor-infiltrated neutrophils produced proangiogenic factors including VEGF and MMP9 in response to IFNβ in mouse melanoma and fibrosarcoma models [245] or could induce tumor cells to elevate VEGF expression [246]. Angiostatin has potent antiangiogenic influence, but understanding of its mechanism is incomplete. Benelli R et al. proposed an interesting theory, such that angiostatin is a strong suppressor of neutrophil migration and its anti-angiogenic effect results from deterrence of neutrophil mobilization into tumors [247]. A study using mutant *K-ras* mouse model of lung cancer showed neutrophil-tumor influx was suppressed upon inhibition of CXCR2 and the diminished neutrophil presence in the tumor was followed by significant tumor reduction [248]. The authors speculated that lack of neutrophil-derived elastase and thus reduction of angiogenesis could be the mechanism of tumor growth inhibition after neutrophil blockade. Still, the role of neutrophil elastase on angiogenesis remains undecided [249].

Chemokines and their receptors mediate protumoral neutrophil or PMN-MDSC recruitment to tumor sites which play critical roles for tumor evasion from antitumor T cells. CXCR2^+^ MDSCs were recruited to CXCL2- and MIF-expressing human bladder tumors and those tumor-associated MDSCs exhibited stronger suppression of T cells than the counterpart cells from the peripheral blood of cancer patients or healthy donors [250]. Indeed, MDSCs recruited to tumors by chemokines can inhibit T cells via MDSC-derived arginase-1 [177,178], but chemokines might have additional important effect on these processes [251]. Arginase-1 is stored within granules of neutrophils and released upon neutrophil activation. In this study, arginase-1 release was induced by non-small cell lung cancer (NSLC) cell culture supernatant, specifically tumor cell-produced IL-8 and TNF. IL-8 silencing reduced the tumor cell capacity to induce arginase-1 release by neutrophils. A recently published study demonstrated cancer–chemokine–neutrophil–T cell inhibition axis working on pancreatic cancer [252]. Here, CXCL1 overexpression in human pancreatic ductal adenocarcinoma (PDAC) was identified using transcriptomic analysis of *KRAS*-*TP53* co-mutated versus *KRAS*-altered/*TP53*^WT^ PDAC cell lines, as *KRAS*-*TP53* co-alteration was associated with worse survival in patients with advanced PDAC. These findings were further confirmed in preclinical mouse models and in patient-derived PDAC tumors. Using snRNA-seq and imaging mass cytometry, CD8^+^ T cells were spatially excluded from the contiguous PanCK^+^ CXCL1^+^ tumor cell and tumor-infiltrated neutrophil (CXCR2^+^ CD11b^+^ CD15^+^) communities. Importantly, silencing CXCL1 in the tumor cells reprogramed neutrophils, controlling tumor growth in a T cell-dependent manner. The authors suggested an interesting mechanism that CXCL1-mediated neutrophil recruitment is the primary signal necessary to rewire the TME in favor of tumor growth.

CXCR1/2 regulation of tumor immune microenvironment can suppress T cell responses in as-yet-unappreciated manners. In an experimental metastasis model by intravascular injection of PyMT breast cancer cells, *CXCR2* knockout (*CXCR2*^fl/fl^::*LysM*^Cre/+^) mice had less tumor metastasis in the lung than control mice, and CD8^+^ T cells from the knockout mouse lungs showed efficient tumoricidal activities compared to the T cells from their control mouse lungs [253]. Systemic CXCR2 antagonist also enhanced the antitumor activity of CD8^+^ T cells. Next, they focused on finding another immune factor that might be playing between CXCR2-driven neutrophils and CD8^+^ T cells. Tumors of *CXCR2* KO mice had elevated CXCL13 levels to recruit B cells and B cell-derived CXCL11 in turn attracted CXCR3^+^ T cells into the tumor. CXCL11 and B cell marker B220 positively correlated with a better clinical outcome of breast cancer, suggesting tumor-associated neutrophils can suppress CD8^+^ T cells by blocking another protective immune branch acting around the tumor sites. Another study revealed CXCR1 and CXCR2 have a novel role in PMN-MDSC inhibition of antitumoral CD8^+^ T cells [254]. While it is a consensus that CXCR2 ligands mobilize CXCR2-expressing neutrophils, neutrophil-attracting chemokines including CXCL1, CXCL2 and CXCL8 directly induced NETosis of human neutrophils. Culture supernatant from tumor cell lines also induced NETosis in CXCR1 and CXCR2 signaling-dependent manners. Inhibition of NETosis sensitized tumors to immune checkpoint therapy. The suggested mechanism behind PMN-MDSC suppression of CD8^+^ T cells in this situation was that PMN-MDSC-derived NETs block CD8^+^ T cells from contacting tumor cells. 

MDSCs can deregulate T cell immunity by modifying components of the trafficking system required for regular T cell migration and homing. Protein nitration is the consequence of local production of reactive nitrogen species (RNS), such as peroxynitrtie anion. Tumor-infiltrated myeloid cells produce RNS, causing tumor sites to become protein nitration-rich compared to neighboring normal tissues [255]. CCL2 nitration in the TME reduces CCL2 capacity to recruit T cells without affecting monocyte attraction, resulting in the selective enrichment of MDSCs in the TME. Separately, MDSCs can interfere with naïve T cell trafficking into and out of lymph nodes. The report by Ku AW et al. revealed that MDSCs could suppress T cell adaptive immunity remotely from tumor sites, but through a contact-dependent manner [71]. MDSCs encounter naive T cells in blood, then induce the T cells to lose L-selectin which is required for T cells to home to lymph nodes. Another study proposed a possibility that MDSCs generated in the spleen of tumor-bearing mice could induce dysregulation of surface molecules important for T cell tissue and lymph node trafficking such as CD44, CD69, and CD62L, thereby inhibiting T cell activation [256].

Chemokines mediate MDSC promotion of cancer metastasis. Cancer cells need to acquire a motile phenotype by random mutation and selection to metastasize. Using a spontaneous murine model of melanoma, infiltration of PMN-MDSCs into primary tumors, mediated by CXCL5, was necessary for the tumor cells to disseminate through EMT [187]. PMN-MDSC coculture led NBT-II bladder carcinoma cells, one type of EMT model cell lines, to acquire mesenchymal phenotypes and to lose desmoplakin junctions (Desmosomes). Desmosomes are adhesive junctions mechanically joining epithelial cells and are essential for tissue integrity. PMN-MDSC-cocultured NBT-II cells migrated further than the NBT-II cells not cocultured. PMN-MDSC coculture also downmodulated E-cadherin expression of a melanoma cell line. In another study, COX-2 in nasopharyngeal carcinoma cells (NPCs) was required for MDSC-promoted metastasis [257]. Tumor COX-2 programed MDSCs to promote NPC migration and invasion by triggering EMT. One of the interesting findings in this study is MDSC induction of tumor cell EMT depended on MDSC direct contact with the tumor cells, although the underlying mechanism remained unaddressed.

Snail is a transcription factor regulating cell adhesion and migration during embryonic development and is a key factor contributing to cancer aggressiveness and metastasis [258]. *Snail* knockdown in mouse ovarian cancer cells slowed tumor growth in immunocompetent mice with a decrease of MDSCs. Snail upregulated expression of CXCR2 ligands to attract MDSCs in a CXCR2-dependent fashion. The mortality of ovarian cancer is largely due to metastatic burden, and Snail, CXCL1/2 levels, and MDSCs were all correlated with poor survival of ovarian cancer patients. It is clear that Snail plays a pivotal role in cancer progression by suppressing E-cadherin to activate EMT, however Snail also promotes tumor aggressiveness by recruitment of MDSCs. Thus, CXCR2 can be an immunological therapeutic target to inhibit progression of Snail-high tumors. 

CXCR2 chemokines are critical for premetastatic niche formation. The study by Wang D et al. using orthotopic mouse model of colorectal cancer indicated that primary colorectal tumors induced CXCR2^+^ MDSCs to accumulate in premetastatic liver where the MDSCs promoted survival of cancer cells undergoing metastasis [259]. The accumulation of MDSCs in the liver was not mediated by liver-derived chemoattractants but by CXCL1 produced by primary tumor-associated macrophages. They suggested primary tumor produced VEGF that activated macrophages to produce CXCL1. It remains undetermined how primary tumor-derived CXCL1, not premetastatic niche-derived factors, induces MDSCs to accumulate in the liver, and whether tumor-derived VEGF is the major driver of macrophage expression of CXCL1 in vivo. Still, this study provided a possibility that CXCL1 could be a promising therapeutic target to alleviate colorectal tumor metastasis.

As mentioned previously, CXCR2 is not exclusively expressed by neutrophils, and may also be expressed by endothelial cells and tumor cells, meaning that CXCR2 could mediate migration of those non-neutrophil cells. Therefore, results from whole body knockout of *CXCR2* or systemic inhibition of CXCR2 and its ligands should be interpreted accordingly. In a study for tumor-stromal interaction in pancreatic tumors, transwell assays showed PDAC cells and CAFs attracted each other in a CXCR2-dependent manner [260]. Additionally, Two papillary thyroid carcinoma cell lines, KTC-1 and B-CPAP cells, expressed CXCR2 and its ligand CXCL5 induced migration of those cells. Furthermore, CXCL5 alone could induce EMT program of PTC cells, including downregulation of E-cadherin and upregulation of vimentin [261]. c-Met is sometimes required for brain metastasis. Brain metastatic variant tumor cells expressed CXCL1 and CXCL8 in a c-Met signaling-dependent manner. CXCL1 and CXCL8 induced tumor cell binding to brain endothelial cells, and endothelial cell tube formation/proliferation [262]. Therefore, targeted deletion of CXCR2 is a more rigorous model. The best neutrophil-specific *CXCR2* KO would be *CXCR2*^fl/fl^::*Ly6G*^Cre/+^ because Ly6G is expressed only in neutrophils in mice [263]. 

As discussed above, TME production of chemokines to recruit and retain myeloid cells is indispensable to tumorigenesis (Figure 5). Tumor cells are the major source of myeloid cell attracting chemokines [211,252,254,262]. Tumor-associated myeloid cells are also the major producers of these chemokines, indicating a positive feed loop for myeloid cell accumulation in tumors [211,264,265,266,267,268]. TME reprograms CAFs to elevate expression of MDSC-attracting factors [252,260,267]. Treg cells might be one of the major sources for CXCL8 [269] or interfere with productive interaction of DCs and cytotoxic T cells by interrupting chemokine-mediated DC-T cell communication [217,270]. Endothelial cells (ECs) also produce CCLs and CXCLs to induce immune cell migration [271,272,273,274]. Tumor endothelial cells (TECs) exhibit unique phenotypes and functions different from normal endothelial cells and sometimes are the central place in the TME to shield tumors from antitumoral immunity. One study which characterized the cellular landscape of human liver from development to disease using scRNA-seq and spatial transcriptomics reported an immunosuppressive mechanism mediated by endothelial cells [272]. This study identified the emergence of PLVAP^+^ ECs and FOLR2^+^ macrophages in HCC both of which were found only in fetus, and showed those ECs and macrophages were colocalized with Tregs in HCC tissues. Tumor hepatocyte-derived VEGFA induces DLL4^+^ PLVAP^+^ tumor ECs, the ECs initiate reprogramming of the macrophages by triggering Notch signaling, and the FOLR2^+^ macrophages expressing a multitude of immunoregulatory chemokines and cytokines recruit Tregs, forming the immunosuppressive environment in HCC. Another study investigated immune mechanisms underlying ovarian cancer metastasis to peritoneal cavity [275]. Notch signaling led endothelial cells to express CXCL2 and the CXCL2 recruited monocytes to metastasized tumors. Interestingly, monocytes required intact Notch signaling in ECs to acquire MDSC phenotypes and functions in the tumor, and EC-derived CXCL2 was a sufficient factor priming monocytes to complete differentiation into MDSCs.

#### 3.1.3. Other Chemokines 

CCL20 is a proinflammatory chemokine that recruits CCR6-positive immune cells, including dendritic cells, T cells, and B cells [276,277]. Its expression is elevated in multiple types of cancer and linked to myeloid cell infiltration, immune evasion, and tumor aggressiveness [278,279]. B16 melanoma, engineered to overexpress CCL20, grew slowly in *CCR6* knockout mice compared to WT mice, with fewer leukocytes that infiltrated into the melanoma in the absence of CCR6. This study indicates the importance of CCL20/CCR6 axis for melanoma growth [280]. Breast tumors recruit CD1a^+^ immature dendritic cells by releasing CCL20 [281,282] and then instruct those immature dendritic cells to prime IL-13 secreting CD4^+^ T cells that facilitate tumor development [281]. Further, CCL20 was identified to be one of the most abundantly expressed chemokines in melanoma by analyzing the secretome of tumor-macrophage coculture [283]. Similarly, colon adenocarcinoma engineered to express CCL20 acquired increased tumorigenicity by inducing intratumoral infiltration of tolerant dendritic cells [284]. Generally, one type of chemokine receptor recognizes multiple different types of chemokine, so blocking one type of chemokine might not be totally effective. Similarly, chemokine receptor blocking might lead to unwanted fallout. CCL20 is the sole high affinity ligand to CCR6, thus CCL20-CCR6 might be targetable with less off-target effects. One major hurdle, however, is that CCR6 is also expressed in other immune cells, especially Th17^+^ T cells. Therefore, cell-specific targeting of the receptor is necessary, but remains a challenging issue.

CCR5 and its ligands are expressed in high levels in some tumor types and is particularly strongly associated with high rates of metastasis and unfavorable outcome of patients with basal or node-negative breast cancer [285]. While the roles of CCR5 and its ligands on T cell effect on tumorigenesis are both positive and negative [125,126,127,133,286], their roles on myeloid cells are almost unanimously protumor. CCL5 is an important chemokine that recruits PMN-MDSCs to tumor sites and promotes the proliferation of CCR5^+^ PMN-MDSCs in the BM, greatly supporting tumor development [287]. A study using an orthotopic 4T1 breast tumor model showed that the autocrine CCL5/CCR5 axis sustained an influx of immunosuppressive myeloid cells into the tumor and discouraged cytotoxic CD8^+^ T cells [288]. Diffuse large B cell lymphoma released CCL5 to attract monocytes into the tumor stroma and these monocytes supported B cell survival and proliferation [155]. CCR5 antagonists and an anti-CCR5 antibody had been developed for HIV treatment and there are attempts to treat cancer by repurposing those reagents. However, like the case of CCR6, cell specific control of CCR5 is necessary to maximize the efficacy of the treatment with their side effects minimized.

CX3CL1 is a chemokine that attracts CX3CR1-expressing cells and has both physiological and pathological roles [289,290]. CX3CL1/CX3CR1 axis has been anticancer or procancer in context-dependent manners [291,292]. Crosstalk between cancer and macrophages via CCR2 and CX3CR1 was an important mechanism that drives lung cancer development and metastasis [293]. Treatment of CT26 colon carcinoma-implanted mice with anti-CX3CR1 antibody in combination with anti-PD-1 antibody enhanced treatment efficacy by remodeling the intratumoral innate immune milieu to be less immunosuppressive [294]. On the contrary, injection of dendritic cells engineered to express CX3CL1 to tumor-bearing mice led to accumulation of antitumoral T cells in the tumor milieu and suppression of tumor growth [295]. The benefit of targeting CX3CR1–CX3CL1 is that they are the only ligand receptor to each other and inhibition of CX3CL1–CX3CR1 signaling can suppress tumor cell migration directly in case that tumor cells express CX3CR1 [296].

### 3.2. Non-Chemokine Mediators to Induce Tumor Infiltration of Myeloid Cells

#### 3.2.1. Complement

The complement system, an important immune surveillance and immune effector mechanism against infection, also has immunomodulatory roles in cancer. Complement factors, C3a and C5a, are potent chemoattractants of innate immune cells [297,298,299], and they are responsible for tumor infiltration of myeloid cells. Tumor production and activation of complement C3 and C5 were necessary for tumor growth in mouse tumor challenge models [300,301]. The study by Zha H et al. found C3 was expressed by various tumor cell lines such as B16F10 melanoma, EL4 T lymphoma, 4T1 breast cancer, CT26 colon cancer, and LLC lung cancer cells [301]. C3-deficient CT26 or LLC tumors had higher numbers of T cells and those T cells were in more activated state than WT tumors because tumor-derived C3a promoted accumulation and immunosuppressive activity of tumor-associated macrophages. On the other hand, tumor-derived C3a induced accumulation of prometastatic immature low-density neutrophils in the liver of mice harboring metastatic lesions [302]. Furthermore, in a TC-1 syngeneic model of cervical cancer in mice, complement fragments were extensively deposited in engrafted tumors [303]. In this study, C3 deficiency or C5aR inhibition impaired tumor growth and tumors lacking C5aR signaling showed minimal infiltration by Gr1^+^ CD11b^+^ MDSCs compared with WT tumors. In addition, MDSCs from C5aR-deficient tumor-bearing mice were less immunosuppressive than MDSCs from WT tumor-bearing mice in terms of inhibition of T cell proliferation and production of ROS and RNS. Because complement products in the TME promote MDSC-mediated immunosuppression in multiple ways [300,301,302,303,304] and C5aR and C3aR are almost exclusively expressed in myeloid cells in the immune system, traditional cancer therapies/anti-PD-1 therapy coupled with blocking complement signaling can be an excellent strategy to treat cancer types where the complement system actively contributes to tumorigenesis. 

#### 3.2.2. VEGF

Anti-VEGF and anti-VEGF receptor (VEGFR) drugs and antibodies are being used for cancer therapy, as it can efficiently control tumor growth by suppressing angiogenesis [305]. However, VEGF can impact cancer in angiogenesis-independent manners, including an autocrine regulation of VEGFR^+^ tumor functions [306]. It is also becoming clear that the traditionally-accepted VEGF–angiogenesis–tumor growth axis contains an additional essential mediator, innate immune cells; those myeloid cells facilitate neovascularization by providing MMPs required for activating angiogenic factors and tissue remodeling or by themselves producing VEGF [307]. On the other hand, VEGF directly or indirectly recruits myeloid cells into inflamed tissues or tumors [308,309,310,311,312]. VEGF-A stimulates lymphangiogenesis and hemangiogenesis in inflammatory neovascularization via macrophage recruitment [313]. VEGF is one of the factors to recruit monocytes and macrophages into hypoxic and necrotic tumor areas [314]. Therefore, therapeutic inhibition of VEGF-VEGFR can control tumors not only by suppressing angiogenesis but also by regulating immune cell recruitment and function within the TME.

#### 3.2.3. Leukotrienes (LTs)

LTs have profound impact on mammalian biology and are also a fundamental regulator of immunity. As potent chemoattractants, LTs induce migration of neutrophils and macrophages during infection and tissue injury [315,316,317,318]. Up-to-date reviews about leukotrienes in immunity are available elsewhere [319,320]. LTs also contribute to tumor-associated inflammation, tumor growth and resistance to immunotherapy [321]. Importantly, retrospective studies and clinical trials showed LT inhibition could reduce the risk from cancer in humans [322,323,324,325]. LTs directly regulate LT receptor (e.g., BLT2)-expressing tumor cells, thus promoting tumor growth and progression [326,327,328]. Furthermore, the effects of LTs on tumors can be mediated by the immune system [329,330,331,332]. The mouse strain with mutant *K-ras* gene (*K-ras*^LA1^) had spontaneously developing lung tumors whereas *K-ras*^LA1^ in the background of *BLT1*^–/–^ showed fewer and smaller tumors [332]. In the same study, crystalline silica (CS) exposure led to increased incidence of lung tumors and this process was attenuated in the absence of functional BLT1. While not changing CXCL and CCL chemokine levels in the tumor, BLT1 deficiency led to significantly reduced recruitment of neutrophils. BLT-1-dependent CS-induced inflammation and tumor growth were further validated using a tumor implantation model. CS-mediated inflammation was multifactorial, but LTB_4_ from macrophages and mast cells and LTB_4_-induced neutrophil accumulation were the central mechanism to chronic inflammatory microenvironment in favor of tumor formation. 

5-Lipoxygenase (5LO) is an enzyme in the arachidonic acid pathway required to produce LTB_4_ and cysteinyl leukotrienes, and it is upregulated in human colon polyps and cancer. In a *APCΔ468* mouse tumor model, *5LO* knockout suppressed intestinal polyposis and rewired the tumor immune milieu [329]. PMN-MDSC infiltration into polyps of *5LO* knockout mice was greatly decreased whereas M-MDSC infiltration was not. Interestingly, 5LO deficiency decreased arginase-1 activity in MDSCs too. CD8^+^ T cell population in the tumor of 5LO-deficient mice did not change, but CD4^+^ T cells including CD4^+^ FoxP3^+^ T cells were significantly decreased. This study proposed that 5LO/LTs-induced conditioning of tumor immune environment is a master regulatory factor to determine colon polyp/tumor development. Innate immune cells not only respond to LTs but also produce LTs in tumors. In a desmoplastic intrahepatic cholangiocarcinoma (ICC) mouse model, cancer-associated fibroblasts (CAFs) recruited CD33^+^ MDSCs to the tumor and also mediated hyperactivation of 5LO metabolism in the MDSCs through CAF-derived IL-6 and IL-33 [330]. ICC tumor cells expressed a LTB_4_ receptor, BLT2, and cancer stem-like cells among the tumor cell population expressed especially higher levels of BLT2. LTB_4_ from CAF-activated CD33^+^ MDSCs stimulated BLT2 signaling in tumor cells, enhancing tumor stemness. In contrast, LTB_4_/BLT can activate antitumor immunity [333,334]. In a TC1 cervical cancer mouse model, *BLT1* KO mice were more vulnerable to the cancer than WT mice [334]. The frequency of tumor-infiltrating CD8^+^ T cells was low in the KO mice compared to WT mice and adoptive transfer of CD8^+^ T cells from tumor-primed WT mice protected the *BLT1* KO mice from the cancer. Therefore, there is the same dilemma as in other cases; LTB_4_/BLT axis toward tumors is contextual. Thus, it is important to be able to pre-determine if this axis is pro- or antitumoral in individual patients before using it as a cancer therapy. 

#### 3.2.4. Damage-Associated Molecular Patterns (DAMPs)

Tumor growth is accompanied by the generation of DAMPs which are released during cell damage and death, and DAMP-induced signaling regulates immune responses, including infiltration of myeloid cells into tumors [335]. High mobility group box 1 (HMGB1) is one of the DAMPs enriched in necrotic tissue sites and TME [336,337]. HMGB1 binds the receptor for advanced glycation end-products (RAGE), Toll-like receptors and TIM-3, and the redox state of HMGB1 determines the activity of the protein. Review articles discussing HMGB1 and cancer are available elsewhere [338]. HMGB1 directly regulates a range of functions of immune cells, including migration [339]. The release of HMGB1 from tumor cells induced recruitment of neutrophils and MDSCs, promoting metastasis [340,341]. Intermittent intense UV exposure is an important etiological factor in the initiation and development of melanoma. A study provided a new mechanistic link from UV-induced DNA damage of epidermal keratinocytes to metastatic melanoma, which was a series of events of UV exposure, HMGB1 release from damaged cells, HMGB1-TLR4-Myd88 signaling-mediated inflammation [340]. In this report, a key mediator of the pathway was HMGB1-mediated recruitment and activation of neutrophils. Expression of RAGE, a receptor for DAMPs, is upregulated in several tumor types [342,343]. RAGE signaling in immune cells drives DMBA/TPA-induced skin tumor development by sustaining myeloid cell infiltration [344]. RAGE also induces protumoral myeloid response for pancreatic carcinogenesis [345,346].

Among the DAMPs, extracellular ATP and adenosine which are in very low levels in healthy tissue and fluid are present in high concentrations under inflammatory, stressed and cancerous conditions [347,348]. Myeloid cells including macrophages and neutrophils are the major immune cell type sensing extracellular nucleotides/nucleosides through purinergic receptors such as P1 and P2 family and accordingly conduct immune functions [349,350]. Extracellular nucleotides/nucleosides are alarming signals; they prepare phagocytes to clear apoptotic cells [351] and promote migration and chemotaxis of myeloid immune cells [352,353,354,355]. Although the role of ATP remains debated [356], adenosine in the TME are immunosuppressive and tumor-supportive. A2A adenosine receptor (A2AR) protected tumors from antitumor T cells [357] and blockade of A2AR suppressed the tumor metastasis by enhancing antitumor NK cell response [358]. Inhibition of two ectoenzymes important for the metabolism of extracellular nucleotides, CD73 (ecto-5’nucleotidase) and CD39 (ectonucleoside triphosphate diphosphohydrolase-1), greatly improved antitumor immune responses, thereby slowing tumor growth [359,360]. CD39 and CD73 were found to be associated with poor survival from ovarian cancer [361]. In this study, immunohistochemical analysis of human ovarian cancer tissues revealed CD39 was expressed on tumor-infiltrated myeloid cells and CD73 was mostly on tumor cells. Ovarian tumor cell-derived adenosine attracted monocytes and those myeloid cells could inhibit CD4^+^ T cells. Another study also showed extracellular ATP (eATP) metabolites regulates tumor immune microenvironment. A new antibody to CD39 was developed which could block CD39 enzyme activity without cell depleting effects and its monotherapy potently suppressed growth of subcutaneously injected tumors in mice [362]. eATP metabolism in the TME is required for tumor-supportive immune environment including increase of immunosuppressive myeloid cells and decrease of intratumor T cell activation, through blocking the tumor-suppressing eATP–P2X7R (an ATP-gated channel)–inflammasome–IL-18 pathway.

PANX-1 is a channel that releases intracellular ATP to extracellular space. While PANX-1 is anti- or protumoral [363,364,365], the role of PANX and PANX-released eATP to modulate tumor immune microenvironment is not definitively decided yet. An interesting molecular model explaining how PANX-1-mediated eATP release from cells could shape inflammation was proposed [366]. PANX-1 is in the same protein complex with P2X7R on plasma membrane of monocyte and macrophage cell lines, ATP released through PANX-1 activates P2X7R in an autocrine manner, then activated P2X7R triggers intracellular signaling to promote IL-1β maturation/release. IL-1β is a pleiotropic proinflammatory cytokine implicated in tumor growth and progression [367]. eATP can be a direct driver of the tumor immune milieu formation by its chemotactic effect of immune cells [353,368,369]. Interestingly, eATP concentration in the TME can also induce qualitative changes of recruited immune cells [370]. 

Purinergic receptors and ectonucleotidases are targetable points to control cancer. As with other targets, purinergic receptors are sensitive to the functional dichotomy of protumoral or antitumoral effects. This is mainly dependent on the cell type that express purinergic receptors (e.g., their expression on tumor cells versus on immune cells) and, in case that immune cells express those receptors, the immune cell type and the immunostimulatory/immunosuppressive state of the immune cells. An in-depth discussion of purinergic receptors in the context of tumors is available elsewhere [371]. Still, it is worth discussing P2X7R a bit more. Immunogenic cell death (ICD) is a cancer cell death strongly, but secondarily, induced by anticancer treatment. ICD results, at least in part, from tumor antigen-specific T cell generation, driven by the massive release of tumor antigens. Alongside tumor antigens, DAMPs are released and DAMP-receptor signaling including eATP-P2X7R fortifies the antitumor immunity [372]. P2X7R stimulation by eATP leads to inflammasome formation which in turn produces functional IL-1β, potently driving inflammation including enhanced chemokine expression [372,373,374]. As mentioned above, P2X7R induces chemotaxis and fast migration of immune cells, thus eATP-P2X7R is highly likely to contribute to ICD by driving immune cell infiltration too. B16 melanoma and CT26 colon cancer grew significantly fast in *P2X7R* knockout mice [375]. In the same investigation, it was established that P2X7R on immune cells was responsible for the P2X7R suppression of tumor growth using BM transfer of WT and *P2X7R* KO. The tumors in *P2X7R* KO mice lacked infiltrated immune cells to an exceptional degree even though this KO mouse strain has an inherent defect in secreting IL-1β. Further analysis showed highly defective migration of *P2X7R*^–/–^ myeloid cells. P2X7R modulation of immunity is linked to tumor promotion as well. P2X7R deficiency restricted lung carcinogenesis with more T cells mobilized and M2-like TAM polarization impaired [376]. P2X7R activity in tumor-infiltrating lymphocytes induces cellular senescence of those T cells and limits tumor suppression [377]. There is a high level of complexity in the roles of extracellular nucleotides/nucleosides and of the proteins metabolizing and sensing them in the tumor immune environment. Therefore, we require a better mechanistic understanding of how each gene involved in extracellular nucleotides/nucleosides signaling modulates tumor immunity, first, tumor type or tumor genotype-specifically, second, with regard to the protein expression on tumor cells versus immune cells.

#### 3.2.5. Platelet and Coagulation

One more mechanism for tumor recruitment of myeloid cells is through platelets or coagulation. Cancer is often thrombogenic [378,379], and platelets and coagulation factors promote cancer sustenance, invasion, and metastasis in multiple ways [380,381,382,383]. One mechanism by which platelets promote cancer metastasis is mediating formation of heteroaggregates comprised of cancer cells, fibrinogen, platelets, and leukocytes; cancer cells require this type of complex interactions with neutrophils, monocytes and platelets in order to bind endothelial cells, activate the microvascular endothelium and promote hematogenous metastasis [194,384,385,386,387]. Platelet-derived signals are required for the rapid recruitment of granulocytes to tumor cells to form “early metastatic niches” which are crucial for metastasis [388]. In this study using a mouse model of tumor metastasis, tumor-platelet-granulocyte microthrombi started forming in the blood vessels of the lungs within two hours after intravascular injection of MC38 colon carcinoma cells. The microaggregate formation was platelet-dependent, based on the observation that platelet depletion greatly reduced the recruitment of granulocytes to the tumor cells. Either platelet depletion or granulocyte depletion led to less metastasis to the lungs. Interestingly, supernatant of platelet-tumor cell coculture contained CXCL2, CXCL5, CXCL7 and activated neutrophil migration. In line, treatment of CXCR2 blocking antibody prevented the formation of the early metastatic niches and subsequent metastasis. 

Tissue factor (TF) is associated with poor survival of patients with some types of cancer, including melanoma. Metastatic melanomas express 1000-fold higher levels of TF than non-metastatic melanoma cells and the growth of pulmonary metastases was significantly inhibited by TF blocking [382]. Another study showed tissue factor expression by tumor cells correlated with metastasis, and recruitment of monocytes/macrophages by tissue factor-mediated coagulation was essential for metastasis [389]. In this study, human melanoma cell line A7 engineered to express TF (A7/TF) induced higher levels of platelet clot formation than its parental cell line. The tumor cell clots recruited CD11b^+^ non-granulocytic myeloid cells. CD11b^+^ cell depletion impeded lung metastasis of intravascularly injected B16 melanoma cells while platelet clot formation around the tumor cells remained in the normal levels. B16 cell intravascular injection into mice bearing B16 subcutaneous tumors led to an increase in survival of injected B16 cells and subsequent metastatic lung nodules. Injection of the naturally occurring anticoagulant, Hirudin, reduced lung colonization of the inoculated melanoma cells in the tumor-bearing mice. Lots of literature [380,381,382,383,384,385,386,387,388,389,390,391,392] conveys that coagulation is an important cancer-promoting factor and myeloid cells recruited by platelets mediate coagulation promotion of cancer dissemination, at least in part. Up-to-date discussion of anticoagulation as a complementary option for cancer therapy and prevention is found elsewhere [393,394]. 

#### 3.2.6. Hypoxia and Acidosis

Hypoxia and low pH in the TME are of major hostile TME conditions to discourage antitumor immunity. While these two factors have fundamental effects in tumor immune escape, hypoxia and acidic condition can induce immunosuppression by regulating immune cell trafficking to tumors. An early study showed ICAM-1, VCAM-1 and E-selectin were expressed on endothelial cells in different levels depending on acidity [395]. In the same study, neutrophils rapidly increased surface levels of β2 integrin and adhered on the endothelium more efficiently at a low pH than at the neutral pH, suggesting the acidic condition of tumors might favor myeloid cell infiltration. The acidic TME could promote myeloid cell migration into tumors through its induction of chemokines. Pancreatic, colonic and prostate cancer cell lines expressed higher levels of IL-8 in pH 6.9 and 6.7 than in pH 7.1 and 7.4 [396]. IL-8 expression was also elevated in human ovarian cancer cells under hypoxic environments [397].

Hypoxia has been illuminated as a microenvironmental factor to fundamentally regulate tumor biology. Hypoxic conditions lead to activation of hypoxia inducible factors (HIFs) and the HIFs induce a plethora of changes in tumor cell gene expression independently or via regulation of canonical inflammatory signaling pathways, including NFκB pathway. Hypoxia, tumor and immunity is discussed in other review articles [398,399,400]. Hypoxia regulates immune responses in and toward TME in various ways, including control of myeloid cell influx into the TME and microlocalization of the immune cells into hypoxic regions. The study by Chiu DK et al. revealed tumor hypoxia–CCL26–MDSC axis in hepatocellular carcinoma (HCC) [401]. Myeloid cells were located near or in hypoxic regions in human HCC tissues and conditioned medium from hepatoma cells under 1% O_2_ was better at recruiting splenic MDSC cells than that under 20% O_2_ in transwell chemotactic assays. Profiling of chemokines produced by a hepatoma cell line under hypoxia versus normoxia identified CCL26 and CCL28. CCL26 expression was directly under control by HIFs and detected in high levels in perinecrotic regions of human HCC tissues. Finally, tumor-derived CCL26 could induce MDSC chemotaxis and recruitment to tumors, and both, blocking of CCL26 receptor and treatment of HIF inhibitor Digoxin, suppressed tumor growth. Another study found tumor hypoxia–CCL8–TAMs [198]. It was spotted that TAMs and CD8^+^ T cells were relocated to hypoxic regions during glioblastoma development in spatiotemporal analysis of human glioblastoma tissues. Tissue-derived factors in the new sub-microenvironment including hypoxia instructed TAMs to be immunosuppressive. The molecular drivers to lead TAMs to the hypoxic areas and to confine them there were CCL8 and IL-1β.

#### 3.2.7. ICAM-1 and VCAM-1

Tumor ICAM-1 and VCAM-1 are a modality for tumor dissemination (Figure 6). ICAM-1 and VCAM-1 are the members of cell adhesion molecule (CAM) family which have critical roles in leukocyte adhesion, motility and trafficking [15]. The cell types expressing these integrin ligands are endothelial cells and immune cells. Interestingly, circulating tumor cells can upregulate ICAM-1 or VCAM-1 expression to recruit and retain myeloid cells on them and utilize those immune cells for metastasis. Myeloid cells can facilitate tumor metastasis in multiple mechanisms. They remodel premetastatic tissue sites for disseminated tumor cells [402]. Neutrophils generate neutrophil extracellular traps (NETs) to trap and to chemoattract tumor cells in distant premetastatic sites [403,404]. Myeloid cells can also promote tumor cell metastasis by contact-dependent manners and the tumor cell-myeloid cell association is mediated by molecular interaction of myeloid cell integrins with CAM molecules on the tumor cell surface. 

ICAM-1 correlates with hepatoma tumor stemness and is required for tumor metastasis [405]. Only a small percentage of cells in Huh7 and Hep3B cell population were ICAM-1-positive, but importantly, the ICAM-1-positive subpopulation expressed higher levels of genes for tumor stemness, including Sox3 and Oct4, and made more hepatospheres in in vitro assays. Furthermore, ICAM-1-positive hepatoma cells had a better ability of generating tumors when injected to nude mice. Hepatoma cells isolated from hepatocarcinoma (HCC) patients also included ICAM-1-positive subpopulations and those ICAM-1^+^ cells were greater at forming tumor spheres. Importantly, HCC patients had ICAM-1-postive hepatocytes in blood and those ICAM-1-positive cells formed tumors in higher efficiencies when injected to nude mice than ICAM-1-negative cells did. Strikingly, patients having more ICAM-1-expressing circulating tumor cells (CTCs) showed shorter disease-free survival periods. This study demonstrated ICAM-1 is a strong indicator of tumor stemness and metastasis, although it remains undecided whether ICAM-1 is merely a coincidental marker for tumor stemness or ICAM-1 on tumors has direct functions for metastasis. In a study using H59 (Lewis lung cancer cell line) liver metastasis in mice, neutrophil depletion abolished colonization of intrasplenically injected H59 cells in the liver and provision of the neutrophil-depleted mice with activated neutrophils normalized the capacity of tumor cell metastasis [406]. Tumor cell binding to hepatic sinusoids is the prerequisite for liver metastasis. Neutrophils play an essential role in this early metastatic process mediating tumor cell adhesion to the sinusoids through direct binding to both endothelial cells and tumor cells. Neutrophil-tumor cell binding was dependent on the molecular interaction of neutrophil Mac-1 integrin with tumor ICAM-1. These results are in line with those from an earlier study, which showed human neutrophils were required for human breast cancer cells (MDA-MB-231) to undergo transendothelial migration in vitro [407]. In agreement, others report that A375-MA2 melanoma cells form micro-clusters with neutrophils inside blood vessels in MAC-1 and ICAM-1-dependent manners [408]. Tumor cells in these clusters extravasated more efficiently than free tumor cells. Further, IL-8 secreted from CTCs was necessary for the cell cluster formation and the tumor cell extravasation. Another interesting study also highlighted the importance of tumor cell-myeloid cell binding for metastasis [409]. In this case, interaction between α4β1 of macrophages and VCAM-1 of tumor cells enhanced cancer metastasis by promoting cancer cell survival through PI3K/AKT pathway activation at the metastatic sites with no effect on lung invasion. 

Monocytes might have the same ability to mediate binding of CTCs to endothelium, thereby facilitating tumor metastasis [410]. On the other hand, tumor VCAM-1 can promote cancer metastasis by recruiting monocytes to metastatic sites [411]. In a study that intended to investigate the mechanisms underlying distant metastasis relapse long after successful removal of primary tumor, which the authors dubbed “transition from dormant micrometastasis to overt macrometastasis”, pre-existing highly metastatic cells were isolated from parental MDA-MB-231 population based on bone metastasis gene signature. Indeed, these cells could metastasize into the bone quickly. In contrast, other clones lacking expression of the bone metastasis gene signature were not able to generate bone metastasis within 100 days after injection. One clone (SCP6) did however develop overt bone metastasis in ~10% of recipient mice after more than 4 months. From the bone metastases, cancer sublines (PDs) were reisolated and most of them metastasized efficiently but one subline (PD2R). PDs and PD2R/SCP6 differentially expressed a group of genes, one of which was VCAM-1. *VCAM-1* knockout abolished the bone metastatic ability of highly metastatic PD cell lines. Further analyses revealed soluble VCAM-1 from PD cells recruited integrin α4β1-positive monocytic osteoclast precursors to the bone and the precursor-derived osteoclasts underwent osteoclastogenesis, a critical process for bone metastasis of cancer.

Tumor cells, especially CTCs and cancer cells in early metastasis, can acquire genetic alterations to express ICAM-1 and VCAM-1, which enable both attraction and retention of myeloid cells. Tumor cells have little or no intrinsic capability to adhere to and transmigrate the blood endothelial layer, thus the CTCs manipulate myeloid immune cells to carry them to metastatic sites. Even so, ICAM-1 and VCAM-1 expression is a double-edged sword for tumor cells because the innate immune system can prevent tumor cells from spreading though physical contact-mediated mechanisms. For example, lungs, the most frequent site of metastasis, house a population of innate immune cells including neutrophils, monocytes, eosinophils that act as gatekeepers in the lung vasculature to block and kill the circulating tumor cells [412,413,414,415]. Importantly, myeloid cell binding to tumor cells underlies this antitumor immune mechanism [412,416,417]. Therefore, the premise for tumor metastasis through ICAM-1/VCAM-1–mediated myeloid cell–CTC clustering is such that the myeloid cell–CTC interaction should be permissive. A group analyzed CTC-immune cell clusters using scRNA-seq [418]. Human and mouse cancer patients had CTC–neutrophil microaggregates in circulation; the CTCs included in the clusters were highly metastatic and proliferative, which is consistent with ICAM-1^+^ CTCs having higher levels of tumor stemness [405]. CTC-associated neutrophils had Arg-1^High^ VEGFA^High^ protumoral phenotypes. As such, CTCs detected in patient blood could be CTCs surviving the host surveillance by choosing permissive neutrophils. Alternatively, some cancer patients may have highly immunosuppressive systemic innate immunity that permits CTCs to survive and spread. 

## 4. Therapeutic Implications

Our working knowledge of how the immune system operates during homeostasis and cancer is a powerful tool used to develop therapeutic alternatives to current standard-of-care treatment strategies in cancer and other diseases. Recent decades have fostered significant growth in this field, particularly with the advent of immunotherapy, including checkpoint blockade, and engineered chimeric antigen receptor (CAR) cells; these and other observations warrant discussion. 

Evidence for immune evasion via VCAM-1 expression has been linked to P3 tumor progression. Interestingly, the lack of VCAM-1 expression has been postulated to be a potential biomarker for positive tumor vaccine response in some types of renal cell carcinoma [60]. Examination of VCAM-1 overexpression in the Oncomine Cancer Profiling Database reveals that most tumors other than renal cell carcinoma do not upregulate VCAM-1. Therefore, while this potential biomarker may be revolutionary for a subset of RCC patients, the breadth of its impact may be quite limited [60,419]. Nevertheless, RCC patients frequently rely on surgery and radiation, and this potential biomarker for positive immunotherapy response would be a great achievement for that subset of VCAM-1^low^ patients. 

Limitations in T cell trafficking to tumors is a major roadblock of CD8^+^ T cell immunotherapy. Hickmann et al. show that in immunologically cold tumors, pharmacologic activation of LFA-1 in conjunction with VLA-4 can improve tumor-specific homing and infiltration of CD8^+^ T cells to promote the antitumor response in a CXCL12-dependent manner, which is further improved in combination with checkpoint blockade [56]. Furthermore, VEGF can downregulate LFA-1 within the tumor and tumor vasculature, resulting in T cell immunosuppression [420]. Thus, concurrent antagonism of VEGF/VEGFR may be beneficial.

In line with the idea that CCL19 supports immune surveillance [80], the administration of intratumoral CCL19 mobilized CD4^+^ and CD8^+^ T cells and DCs at the tumor. This was followed by increased levels of antitumor factors including IFNγ, CXCL9/10 and IL-12, and a reduction of immunosuppressive molecules [421]. This study supports the use of immunotherapy to kickstart the endogenous immune system and mitigate immunosuppressive programs. Furthermore, CAR T cells can be engineered to coexpress various chemokine receptors to improve tumor-specific homing and infiltration [422]. This represents opportunities for cellular reengineering to improve existing strategies to promote the antitumor response. Additionally, CAR T cells that coexpress chemokines such as IL-7 and CCL21 exhibit superior expansion without lymphodepleting chemotherapy, and improved migration to tumors, which is conserved among solid tumor types [423]. Similarly, CAR T cells that coexpress IL-7 and CCL19 enhanced the antitumor potential of CAR T cells by activating bystander T cells and forming robust memory populations [424]. These data support the current trajectory of CAR T therapy that suggests CAR T cell efficiency can be improved via coexpression of ligands and receptors required for T cell tracking, like chemokine and cytokine receptors. Particularly, IL-7 and CCL19/21 support the maintenance of T cell zones within lymphoid tissues, which not only drives recruitment of DCs, but also facilitates recruitment of bystander T cells, further promoting antitumor immunity. Thus, these armored CARs hold great promise for cancer patients and have been thoroughly reviewed elsewhere [425]. Furthermore, CCL19 administration alone in a model of colorectal cancer disrupted angiogenic programs, resulting in decreased tumor size and metastasis [426]. The authors speculate that this could be due to the downregulation of VEGF-A and HIF-1α and suppressive microRNA activation. This study further corroborates the benefit incurred from strengthening CCL19/CCL21-CCR7 axis for therapeutic efficacy. 

The recent spatial transcriptomic analysis of human lung cancer patients indicates that spatially organized ‘immunity hubs’ consisting of stem-like IFNγ^+^ T cells in residence with myeloid cells in a CXCL10/11-CXCR3 dependent manner are associated with positive response to anti-PD-1 therapy [101]. This is likely a function of improved activation and recruitment of effector CD8^+^ T cells. IFNγ induces expression of all three CXCR3 ligands, thereby playing a central role in antiviral or pathogenic T cell immunity [95]. Likewise, IFNγ–CXCL9/10/11–CXCR3 axis is significant for cancer immunity. The TME of human ovarian tumors secretes CXCL9 in an IFNγ-dependent manner and thus recruits tumor-reactive T cells, conferring longer survival of patients and better response to checkpoint immunotherapy [427]. CXCL9 and CXCL10 mediated accumulation of CD8^+^ T cells in tumors occurs following checkpoint blockade and these IFNγ-dependent events are mediated by tumor-associated macrophages [428]. In a lung cancer model, radiation has been shown to enhance CXCR3^+^ T cell activation in an IFNγ-dependent manner, via CXCL10 and ICAM-1 [429]. Intratumoral CXCR3 chemokine signaling contributes to anti-PD-1 response rates [430]. Therefore, modulation of CXCR3 ligands can be a potential therapeutic avenue to treat tumors [431,432].

In a human xenograft model of T cell acute lymphoblastic leukemia, CXCR4 antagonism resulted in disease suppression, which may be related to Myc signaling downstream of CXCR4 [433]. Similarly, the blockade of both CXCR4 and PD-1 in a murine model of ovarian cancer revealed increased CD4^+^ and CD8^+^ T cell tumor infiltration [434].

A study with multiple solid tumor types used single-cell meta-analysis to reveal a population of tumor-reactive CXCL13^+^ CD8^+^ T cells within tumors that was associated with improved response to immune checkpoint blockade [435]. Similarly, a recent report of human advanced gastric cancer and anti-PD-1 plus chemotherapy suggested that the addition of Pembrolizumab (αPD-1) to 5-FU/oxaliplatin chemotherapy enhances T cell antitumor immune remodeling, especially CXCL13^+^ CD8^+^ T cells which covaries with multiple tumor-reactive phenotypes [436]. Further, high-grade serous ovarian cancer that expressed high levels of CXCL13 recruited CD20^+^ B cells and CXCR5^+^ CD8^+^ T cells, for overall improved patient survival. This was even further supported in combination with PD-1 blockade [437]. However, in renal cell carcinoma, CXCL13^+^ CD8^+^ T cells were accompanied by immunosuppressive Th2 T cells and exhibited decreased effector function and increased exhaustion phenotypes [438]. Therefore, the CXCL13/CXCR5 axis supports immune cell recruitment and organization within the TME. Current data largely suggest that this axis may be exploited to improve the antitumor response, especially in combination with immune checkpoint blockade [116,117], though there are some reports of opposite effects.

Interestingly, it appears that CD4^+^ T cell tumor infiltration can be important for effective immunotherapy. A study by Huffman AP et al. used scRNA-seq to examine the tumor immune microenvironment following treatment of murine PDAC-derived tumors with agonist CD40 antibody plus or minus immune checkpoint blockade [122]. It showed non-MDSC tumor-resident myeloid populations produced CCL5 in response to CD40 activation, but not in response to immune checkpoint blockade, and CCL5-recruitment of CD4^+^ T cells mediated treatment efficacy. CD4^+^ T cell-centric approaches could provide significant therapeutic benefit alongside approaches to improve cytotoxic T cell activities. 

It is important to discuss the implication of exhausted T cells within the TME. One main function of immune checkpoint molecules is to curb overactive inflammatory responses and dampen the aberrant activity of the immune system that may lead to unchecked inflammation. Tumors can hijack this control mechanism to facilitate tumor growth and survival. As tumors recruit and reprogram myeloid cells to become immunosuppressive within the TME, the tumor infiltrated MDSCs upregulate negative immune regulators such as PD-L1 to induce T cell exhaustion within the tumor. James Allison pioneered immune checkpoint blockade, which blocks these receptors or their ligands to prevent downstream signaling, reinvigorating the immune system. This strategy has shown immense success in treating tumors when combined with existing treatment strategies [439,440]. However, T cell exhaustion phenotypes are more complicated than initially thought, with different populations that arise within the tumor, including precursor-exhausted and terminally exhausted cells. Some exhaustion markers such as PD-1, are coincidentally also markers associated with T cell activation. Sailer C et al. showed that PD-1^high^ CAR T cells impart more robust antitumor immunity than their PD-1^low^ counterparts [441]. This study and others encourage additional investigation into exhaustion phenotypes and function and hold promise for new therapeutic approaches.

Myeloid cells play pivotal roles in every step of the initiation, progression and metastasis of cancer. Based on this, current research to improve cancer treatment efficacy has been focused on targeting and modulating tumor-supportive myeloid cells. Here, we discuss some of the approaches that are used to control these myeloid cells. Restriction of myeloid cell generation and differentiation can be achieved via blockade of growth factor/growth factor receptor signaling (e.g., CSF1R). Treatments targeting tumor hypoxia are under active investigation. Hypoxia-mediated therapies can also offer the benefit of controlling immunosuppressive TME, as immune responses are at least partly responsible for mediating the hypoxia-to-cancer relationship. Additionally, adjuvant or neoadjuvant therapies used in conjunction with conventional strategies can kickstart the endogenous immune response to promote a more robust anticancer response. However, the most precise and effective method would be to target myeloid cell migration and trafficking directly. 

The CCL2/CCR2 axis is crucial for monocyte mobilization into the TME, and a myriad of preclinical mouse tumor models have proven this axis is both protumoral and targetable [180,203,210,213,215,442,443,444]. Additionally, these studies show suppression of CCL2/CCR2 improves anticancer therapies using PD-1 blockade or adoptive T cell transfer [213,445,446,447,448]. One major drawback is the low cell-specificity of CCL2/CCR2 targeting. Antitumoral cytotoxic CD8^+^ cell responses can be mediated by CCR2, therefore, CCL2/CCR2 antagonism can block CCR2^+^ CD8^+^ T cell responses [449] or CCR2^+^ DC-mediated antitumoral immune responses following chemotherapy [450]. CCR2 is also expressed by activated endothelial cells and can induce endothelial cell migration for wound injury repair [451]. Another point to consider is the compensatory accumulation of immunosuppressive neutrophils as a result of inhibition of monocyte recruitment [212]. As such, CCL2/CCR2 targeting may not be a straightforward therapeutic target. 

Neutrophils often constitute the majority of myeloid cells in the TME and, like monocytes, extensive literature points to their tumor-supporting roles [228,231,232,233,234,235,236,237,452,453,454]. Neutrophils in the bloodstream and the TME almost exclusively express CXCR2, which is critical for neutrophil mobilization. Thus, targeting CXCR2 is the most specific strategy to control tumorigenic neutrophil activity. In preclinical animal models, CXCR2 antagonism or dual antagonism of CXCR2 and CXCR1 (CXCR1: a CXCL8 binding chemokine receptor highly expressed on neutrophils) has emerged as a promising complementary therapy potentiating cancer treatments by immune checkpoint blockade and antitumor T cell transfer [250,455,456,457,458,459]. Even more specific strategies would include subtype-specific targeting of neutrophils or intervention of neutrophil-specific activities such as NET formation. Importantly, therapeutic resistance to immune checkpoint inhibitor treatment is partly connected to myeloid cells. As such, combination therapy of CXCR1/CXCR2 antagonism and anti-PD-1/anti-CTLA-4 is under clinical trial [219]. Still, the myeloid cell connection seems to be only part of the resistance to immune checkpoint therapy. In effect, the key to success of this immunotherapeutic strategy would be our ability to predict and select tumor types or patient types that will respond to the combinational therapy before starting the treatment. 

As outlined throughout this text, a better understanding of tumor immune micro- and macroenvironment, development of efficient immunotherapeutic tools and investigation into promising biomarkers are important, active areas of research. Finally, personalization of cancer immunotherapy goes a long way to attain more controlled management of cancer as it is becoming clear that tumors are highly heterogenous and each tumor type in different genetic and environmental backgrounds of individual humans shows different susceptibility and responsiveness to given immunotherapies like any other cancer therapies ever developed.

## 5. Conclusions

Immune cells are highly capable of cellular movement, which is indispensable to an effective immune system. Motility and homeostatic migration of immune cells enable constant immune surveillance of the body to detect infection and aberrant cellular behavior, which is crucial for the early detection of immune insult. During an inflammatory immune response, immune cells are mobilized to their target site. Coordinated expression of adhesion molecules and chemotactic signals ensure that immune cells are activated at the right time and recruited to the right place for successful clearance of infection or disease. Dysregulation of immune cell migration, sensing, or communication can have devastating ramifications for the host including the inability to clear infections and cancer. In cancer, especially the tumor microenvironment, canonical factors may operate differently compared to healthy tissues. Alternatively, new cellular subsets or functions may arise from the complex milieu, such as exhausted immune cells and immune cells that acquire tumor-promoting phenotypes. Additionally, complex cellular interactions within the tumor microenvironment can cause traditional signaling axes to become multi-functional, or to adopt noncanonical pathways (Table 1).

Based on the consensus opinion that cancer is a result of immune evasion, efforts have been made to improve cancer treatment by restoring the antitumor immunity that may become dysregulated in cancer patients. As we outline in this text, some of the efforts have finally paid off. Immune checkpoint therapy has been established as an effective treatment when used alone, or in combination with chemotherapies for about 50 cancer types [460,461]. CAR T cell therapy has become a mainstream treatment method for blood cancers since its first approval in 2017 [462]. Now, research efforts aim to apply CAR T cell therapy to treat solid tumors [463,464]. Activation of neoantigen-specific T cells has been suggested as a promising tumor immunotherapy, with some clinical trials reported so far [465] (clinical trials reported by ClinicalTrials.gov: NCT05292859, NCT05194735, NCT04520711). Another avenue with the potential to aid in cancer treatment is the control and trafficking of immune cells to cancer, including MDSCs and Tregs, summarized in this review. Preclinical and clinical attempts to suppress cancer by antagonizing CCR2, CCR4, CCR5, CXCL8, CXCR1/2, CXCR4 combined with or without other therapies [206,222,229,466,467,468,469,470,471,472,473,474,475,476,477] (clinical trials reported by ClinicalTrials.gov: NCT03177187, NCT03851237), by chemokine modulation regimen via cytokine and TLR agonist/antagonist injection [229] (clinical trials reported by ClinicalTrials.gov: NCT06149481, NCT05570825, NCT03161431), or by targeting non-chemokine/chemokine receptor mechanisms such as eATP metabolism [359,362] (clinical trials reported by ClinicalTrials.gov: NCT05234853, NCT05177770, NCT05431270) or complement [304,478] (clinical trials reported by ClinicalTrials.gov: NCT02257528) are being continued. 

Indeed, cancer poses new challenges to the immune system as it has many mechanisms of evading antitumor immunity. Most important, is the distinct ability of cancer to dynamically alter cell surface proteins that control immune cell migration, signaling, and effector function. Likewise, the expansive immune behaviors that arise as leukocytes attempt to clear malignancies are fascinating. This complex and highly adaptive cellular interplay of the antitumor immune response and cancer immune evasion or immunosuppression represents a unique immunological phenomenon. We and others are encouraged by the recent scientific advances that have strengthened our understanding of tumor-immune interactions. Still, many questions remain unanswered. Moving forward, therapeutics designed to promote tumor clearance or disease prevention should consider the broader landscape, including cellular trafficking, localization, and immune interactions within tissues and the TME. Extensive ongoing research efforts are dedicated to better understand these complexities and to refine therapeutic targets. Our review aims to shed light on some potential paths that can be exploited to promote the antitumor response and support tumor clearance.
cells-13-00844-t001_Table 1Table 1Key chemokines/chemokine receptors and molecules for immune cell adhesion; pro- and antitumor functions.CancerFunctionsOutcome/Mechanism**Cell adhesion molecules, Receptors**


ICAM-1/ICAM-2/ICAM-3, αLβ2 or αMβ2Colorectal cancer,Melanoma,Breast cancer,Pediatric osteosarcoma,Pancreatic cancer,Gastric cancerT cell migration and adhesion**Antitumor**/recruitment of antitumoral T cells to tumor microenvironment (TME) [12,13,14,15,53,54,55,56,63,64,65]Neutrophil/PMN-MDSC migration and adhesion**Protumor**/recruitment of immunosuppressive neutrophils/PMN-MDSCs to TME [16,25,26,27,28], metastasis [406,407,408]**Antitumor**/recruitment and retention of tumoricidal neutrophils to/in TME [412,413]Monocyte/macrophage/M-MDSC migration and adhesion**Protumor**/recruitment and retention of immunosuppressive monocytes/macrophages/M-MDSCs to/in TME [16,29]**Antitumor**/recruitment and retention of antitumor monocytes/macrophages to/in TME [413,414]VCAM-1, α4β1 Renal cell carcinoma,Breast cancerT cell migration**Protumor**/disruption of T cell binding to tumor cells [60]Monocyte/macrophage adhesion/retention to tumor cells**Protumor**/tumor survival at metastatic sites [409], bone metastasis by osteoclastogenesis [411]Selectins, Selectin ligandsBreast cancer,MelanomaT cell trafficking to lymph nodes (LNs) and tumors,Tumor cell interaction with endothelium**Antitumor**/generation of antitumor T cells in LNs [66], T cell infiltration into tumors [73,182]**Protumor**/suppression of T cell generation in LNs by L-selectin shedding/downregulation of T cells [70,71,72], Tumor cell extravasation [67,68]**Chemokines**, **Chemokine receptors**


CCL19/21, CCR7 or CXCR7Lung carcinoma,Breast cancer,Cervical cancer,Gastric cancerT cell migration to/in LNs and TME**Protumor**/metastasis of CCR7^+^ tumors [84,85,86,87], Treg migration to TME [89], T cell sequestration by TME remodeling [91]**Antitumor**/generation and tumor infiltration of cytotoxic T cells [38,81], chemokine scavenge by CXCR7 [119]CXCL9/10/11, CXCR3 or CXCR7Colorectal carcinoma,Melanoma, Fibrosarcoma, Ovarian cancerT cell migration to TME**Protumor**/chemokine scavenge by CXCR7 [104,105], tumor infiltration of CXCR3^+^ Treg cells [106]**Antitumor**/activation and tumor infiltration of CXCR3^+^ T cells [94,95,96,97,98,99,100,101,102]CXCL12, CXCR4 or CXCR7Melanoma,Hepatocellular carcinoma,Ovarian cancerT cell migration and localization to/in TME**Protumor**/sequestration of T cells in TME [107], inhibition of tumor infiltration of T cells [108], recruitment of Treg cells and TAMs [106,110], chemokine scavenge by CXCR7 [120]**Antitumor**/tumor infiltration of T cells [109], retention and microlocalization of CD8^+^ T cells in tumors [111]CCL2, CCR2Lung carcinoma,Melanoma, Glioma,Inflammatory breast cancerMonocyte/macrophage mobilization and localization to/in TME**Protumor**/suppression of CD8^+^ T cells, and promotion of neovascularization and metastasis by M-MDSCs recruited to tumors and premetastatic sites [192,199,210,211,212,213,214,215,216,217,442,443,444,445,475] **Antitumor**/Tumor entrainment of neutrophils – tumor killing [412], antitumor monocyte recruitment to premetastatic lung [413], Recruitment of CCR2^+^ T cells and antigen presenting cells [449,450]CXCL1/2/5/8, CXCR2Melanoma, Gastric cancer, Lung cancer, Fibrosarcoma,Papilloma, Colon cancerNeutrophil mobilization and localization to/in TME and CTCs**Protumor**/suppression of CD8^+^ T cell activities and tissue infiltration, suppression of tumor cell senescence, promotion of tumor genomic instability, neovascularization, invasion, metastasis, and EMT by neutrophils/PMN-MDSCs recruited to tumors and premetastatic sites [187,192,202,212,229,230,231,232,233,234,235,236,237,238,239,240,241,242,243,244,245,246,247,248,249,250,251,252,253,254,255,256,257,258,259,467,470,475], CTC-myeloid cell cluster [388,408]

## Figures and Tables

**Figure 1 cells-13-00844-f001:**
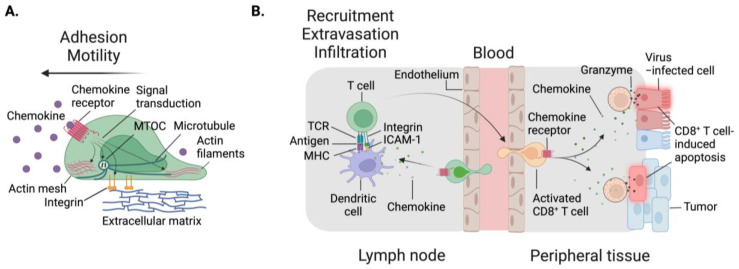
Cellular migration is dependent on cell intrinsic factors, as well as multiple signal inputs from the environment. (**A**) External signals such as chemokines are sensed by membrane-bound receptors on immune cells. The receptor-induced intracellular signaling reorganizes the actin cytoskeleton and the microtubule organizing center (MTOC) and activates integrins, resulting in adhesion to extracellular matrix or counterpart cells, followed by directional migration. (**B**) CD8^+^ T cells migrate to lymph nodes where they are primed by tumor- or virus-antigens presented by dendritic cells. These activated T cells are then recruited to cancerous or infected tissues and eliminate transformed or infected cells by secreting cytotoxic molecules including Granzymes. Immune cells have remarkable capabilities to adapt their mode of adhesion and migration to their surroundings. TCR, T cell receptor; MHC, Major histocompatibility complex.

**Figure 2 cells-13-00844-f002:**
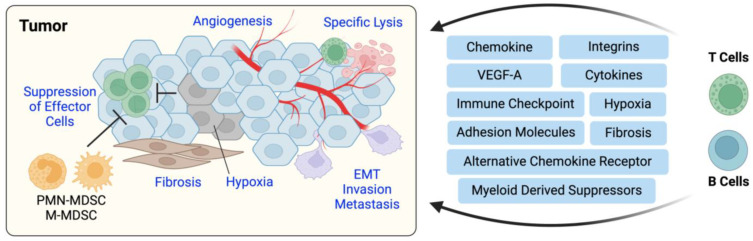
Tumor recruitment of adaptive immune cells can have positive and negative effects to the host. The main effectors, T cells and B cells, are attracted via chemokines, cytokines, and growth factors, which continue to modulate their activity within the tumor microenvironment. Chemokines, adhesion molecules, checkpoint molecules contribute to specific lysis or immune evasion contextually. Recruited immune cells or tumor cells can secrete VEGF-A to further modulate immune cell recruitment. Cancer-associated fibroblasts can modulate the extracellular matrix to further augment effector cell migration within the microenvironment. Tumors become hypoxic, limiting effector capability. All of these factors may contribute to tumor invasion and metastasis. EMT, Epithelial-mesenchymal transition; PMN-MDSC, neutrophil-derived myeloid-derived suppressor cell. M-MDSC, monocyte/macrophage-derived myeloid-derived suppressor cell.

**Figure 3 cells-13-00844-f003:**
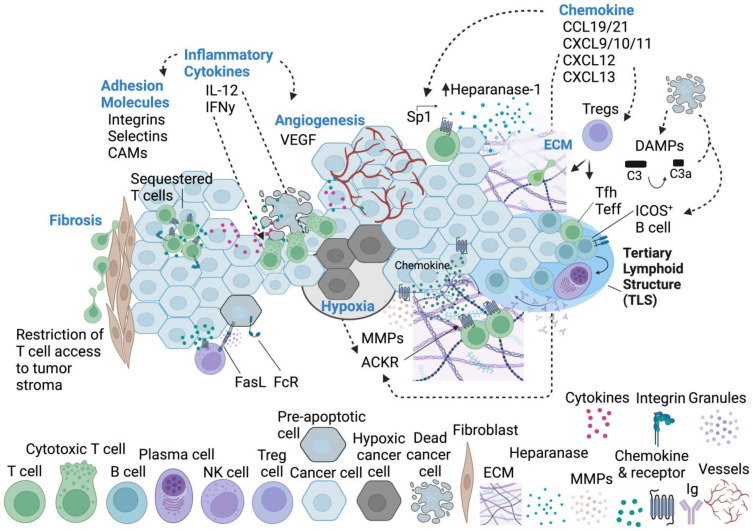
Lymphoid cell recruitment to tumors by multiple mechanisms. Immune cell migration through ECM, tissue, blood, and lymphatics is highly dependent on expression of integrins, selectins, and cell adhesion molecules such as ICAM family ligands. Chemoattractants produced within the TME recruit and retain lymphocytes, which play a significant role in tumor elimination. Chemokine receptor expression on immune cells and tumor cells is highly dynamic. Tumors may alter chemokine receptor and cell adhesion factor expression to sequester or restrict T cell access within tumor stroma, limiting their effector capability. Inflammatory cytokines from immune cells or tumor/tumor associated cells can induce additional alterations to these cell surface molecules. Furthermore, inflammatory cytokines can promote T cell cytotoxicity, causing tumor cell apoptosis. These cytokines can also promote angiogenic programs, increasing flow of nutrients to the tumor, as well as myeloid derived suppressor cells (not pictured). Angiogenic programs can result in aberrant vasculature patterns, resulting in hypoxic tumor regions with limited effector cell presence. TME factors can also induce the expression of atypical chemokine receptors, which scavenge free ligands, disrupting intratumoral immune cell signaling and activation, as there is less bioavailable chemokine for typical receptors. Chemokine and cytokine expression patterns can also cause alternations to the ECM through induction of MMPs or heparanase-1, which can both promote or disrupt immune cell migration depending on the context. Chemokines also recruit Treg cells to the tumor which can induce immunosuppression. DAMPs from apoptotic cancer cells can activate complement signaling cascades and recruit B cells. Interactions between Tfh and B cells promote tertiary lymphoid structure formation, which supports the antitumor response. Furthermore, within tertiary lymphoid structures, interactions with antigen, B cells, and Tfh give rise to plasma cells, which produce large amounts of Igs. NK cells also contribute to immunosurveillance within the TME through recognition of Fas Ligand (FasL), Fc receptors, or TRAIL, though tumor cells have evolved mechanisms to downregulate those receptors to evade antitumor immune surveillance. Fibroblast remodeling contributes to tumor fibrosis, and may disrupt immune cell migration patterns or tumor access. Tfh, T follicular helper cell; Teff, effector T cell; Tregs, regulatory T cells; MMPs, matrix metalloproteases; DAMPs, damage-associated molecular patterns; Ig, Immunoglobulin.

**Figure 4 cells-13-00844-f004:**
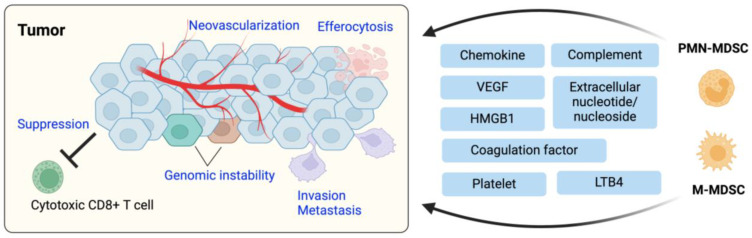
Tumor recruitment of innate immune cells is essential for tumor development, invasion and metastasis. Tumor cells or tumor-associated cells attract and retain protumoral innate immune cells by secreting chemokines, complement, leukotrienes (e.g., LTB4), and damage-associated molecular patterns (e.g., adenosine and HMGB1) or via platelets as a bridge. Recruited myeloid cells support tumorigenesis by suppressing antitumoral T cell response, promoting angiogenesis and genomic instability, clearing dying cells via efferocytosis, and by facilitating tumor invasion and metastasis to neighboring and remote tissue sites. PMN-MDSC, neutrophil-derived myeloid-derived suppressor cell; M-MDSC, monocyte/macrophage-derived myeloid derived suppressor cell.

**Figure 5 cells-13-00844-f005:**
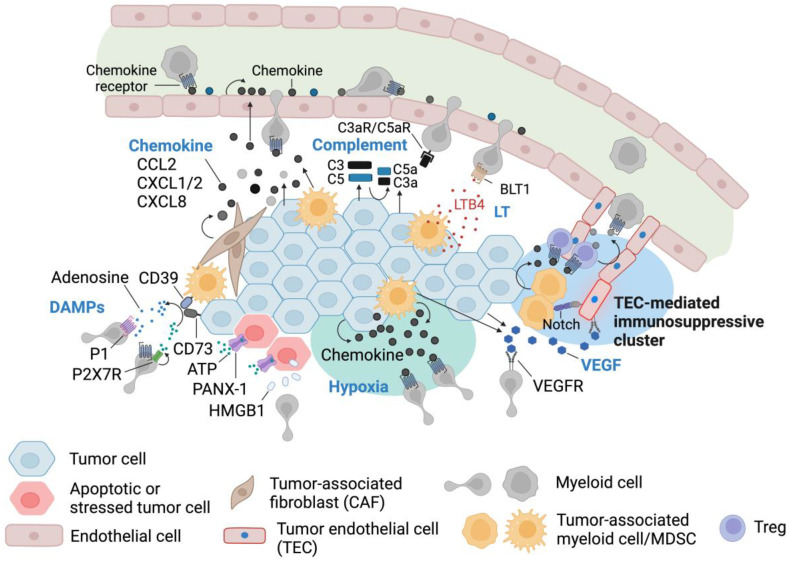
Myeloid cell recruitment to tumors by multiple mechanisms. Chemoattractants produced within the TME recruit and retain myeloid cells, which are indispensable for tumor growth and development. Tumor cells, tumor-associated macrophages (TAMs), and cancer-associated fibroblasts (CAFs) secrete myeloid cell-attracting chemokines such as CCL2, CXCL1/2, and CXCL8. Recruited myeloid cells are reprogramed to support the entirety of tumorigenesis, from tumor initiation and growth to invasion and metastasis. Endothelial cells also produce chemokines and/or display them on glycocalyx that they have on the luminal side. Some tumor types express complement C3 and C5. Their cleaved forms, C3a and C5a, are potent chemoattractants of myeloid cells which express receptors for C3a and C5a (C3aR and C5aR). Leukotrienes (LTs) are critical signaling mediators in mammalian biology. They also play significant roles in the immune system, including cancer immunity. While leukotriene-metabolism is present in a range of cell types, myeloid cells are the major producer of leukotrienes in the TME. Some cancer cells can also generate a large amount of LTs. LT receptors such as BLT1 are expressed broadly in myeloid immune cells and LT binding to their receptors induces adhesion and chemotactic migration of neutrophils, monocytes and macrophages. Hypoxic conditions are inevitable in solid tumors and are an obstacle for growing tumors. However, successful tumors adapt themselves to hypoxia, inducing neovascularization and using it as a tool for immune evasion. Tumor cells and TAMs in hypoxic conditions attract and retain myeloid immune cells by enhanced expression of chemokines, then those recruited myeloid cells facilitate angiogenesis by providing proangiogenic growth factors and MMPs. Myeloid cells under hypoxia are reprogramed toward an immunosuppressive phenotype through anti-inflammatory cytokines including IL-10. VEGF, a critical factor for angiogenesis which is elevated significantly under hypoxia, is also a myeloid cell attractant. TME also has high levels of damage-associated molecular patterns (DAMPs) including High mobility group box 1 (HMGB1) and extracellular nucleotides/nucleosides. DAMPs are secreted passively upon cell death or actively released under stress conditions. Secreted HMGB1 has multiple inflammatory functions including immune cell recruitment. Tissues under inflammatory and cancerous conditions are rich in ATP, which is actively released from cells through specialized plasma membrane channels such as PANX-1. ATP is a strong stimulator of both fast cell migration (mediated by PANX-1/P2X7R complex) and direct chemotaxis of immune cells. ATP can be rapidly metabolized to adenosine by ectonucleotidases including CD39 and CD73, then adenosine induces chemotactic migration and immunosuppression of innate immune cells. Tumor endothelial cells (TECs) are unique structurally and functionally. In hepatocellular carcinoma, VEGF from the TME generates TECs and the TECs are an immune-signaling hub that recruits MDSCs and Tregs.

**Figure 6 cells-13-00844-f006:**
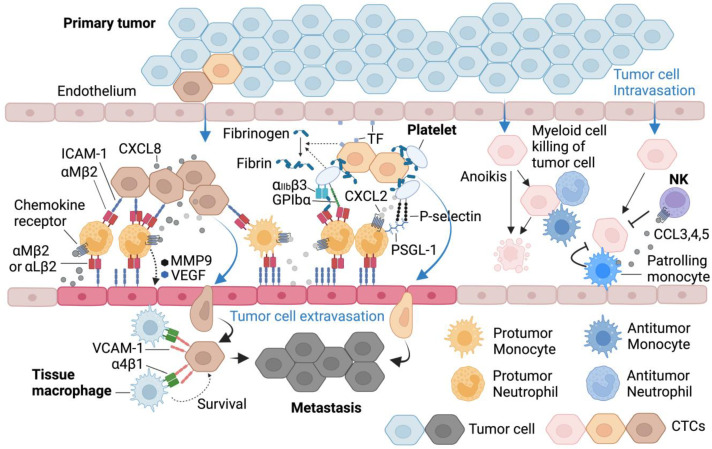
Myeloid cell recruitment to circulating tumor cells (CTCs). CTCs are cancer cells that escape their primary site into the bloodstream, becoming seeds of metastasis. CTCs utilize immune cells, especially myeloid cells and platelets, to survive and disseminate into distant tissues. CTCs can express ICAM-1, and thus recruit and retain MAC-1 (αMβ2 integrin)-expressing neutrophils. Those neutrophils adhere to ICAM-1-expressing endothelium at the same time, thus guiding CTCs to extravasate into distant tissue sites. CXCL8 from CTCs help retain and activate neutrophils in CTC-neutrophil microaggregates while the neutrophils precondition endothelium for tumor cell extravasation by secreting VEGF and MMPs. Monocytes might have a similar process for CTCs to adhere to and extravasate endothelium. CTCs can interact with myeloid cells via platelets. CTC-platelet complex binds to neutrophils via the interaction between molecules on platelets and neutrophils, such as P-Selectin/PSGL-1, GPIbα/MAC-1, αIIbβ3/fibrinogen/MAC-1. Platelets further sustain the ternary complex by producing chemokines to recruit, retain, and activate neutrophils, and by facilitating coagulation in the complex. Tissue factor (TF) derived from CTCs or primary tumor sites drives the CTC-platelet-myeloid cell complex formation. Metastatic tumor cells can express VCAM-1 and tissue-resident α4β1^+^ macrophages promote survival of the VCAM-1^+^ tumor cells in the metastatic sites. Physical association of these two cells via α4β1–VCAM-1 interaction underlies the tumor cell survival. However, humans and mice have antitumoral innate immune mechanisms in blood and in the premetastatic sites to prevent tumor cell spread. Neutrophils, eosinophils, and monocytes block tumor cells in premetastatic sites by directly killing them or via recruiting cytotoxic lymphocytes and NK cells.

## Data Availability

No new data were created or analyzed in this study. Data sharing does not apply to this article.

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
