# Peer review of "Immune Cell Migration to Cancer"

_cells, 2024, doi:10.3390/cells13100844_

Round 1

Reviewer 1 Report

Comments and Suggestions for Authors

Overall, the review presents valuable insights into migratory patterns of immune cells with some of them being closely related to cancer development and metastasis. However, in certain instances, the content appears somewhat broad and vague, making it challenging for readers to follow the discussion cohesively. Furthermore, while the chosen topic is undeniably relevant to scientific community, it might benefit from a more focused approach.

Specifically, I would recommend considering a narrower scope for the manuscript. Perhaps delving into a specific migratory pattern or focusing on a particular cell type could offer a more detailed and comprehensive analysis. By narrowing down the focus, the authors could provide deeper insights and a more cohesive narrative.

Additionally, I suggest dedicating a separate chapter to explore the therapeutic implications of migratory patterns in cancer. This would allow for a more thorough examination of this crucial aspect and ensure that it receives the attention it deserves within the manuscript.

Furthermore, the inclusion of tables to supplement the text would greatly enhance the readability and flow of the manuscript. Tables could be utilized to summarize key findings or reference original studies, providing readers with a visual aid to navigate the complex information presented.

In conclusion, while the review demonstrates significant potential, I believe that implementing these suggestions would strengthen the manuscript and enhance its impact within the scientific community.

Reviewer 2 Report

Comments and Suggestions for Authors

In this review, the authors aim to provide a comprehensive exploration of the advances and knowledge in the literature concerning the migration of innate and adaptive immune cells to cancer. They introduce the fundamental concept of cell movement and migration, delving into both cell intrinsic mechanisms and their responses to external cues. When discussing immune cell migration to tumors, they cover a broad spectrum of cells, including T cells, B cells, NK cells, myeloid, and lymphoid cells. Additionally, they highlight factors influencing immune cell infiltration into cancers, which encompass chemokines, integrins, cell adhesion molecules, chemokine receptors, and even non-chemokine mediators such as complement factors (C3a and C5a), vascular endothelial growth factor-A (VEGF-A), semaphorin (SEMA), and leukotrienes. Despite providing comprehensive literature coverage, there are major and minor concerns outlined below. A major revision is recommended to potentially achieve acceptance.

Major concerns:

1.        The overall logical flow of the main content is poorly organized and confusing, with a mix of topics. At times, innate and adaptive cells are presented as the focus, while suddenly transitioning to discussing chemokines, integrins, and cytokine receptors as key topics.

2.        The authors primarily list facts from publications without delving into in-depth discussion or summarizing diverse results from the literature to propose their own ideas and hypotheses.

3.        Figure 2 attempts to illustrate the key chemokines that directly guide immune cell migration in solid tumors; however, the figure is overly simplified and lacks explanatory depth, providing little assistance for readers in understanding the underlying mechanisms.

4.        Similarly, Figure 3 appears very similar to Figure 2, with limited instructive information and an excessive level of simplification.

Minor concerns:

1.        Figure 1B: “CD8+ T cells migrate to lymph nodes where they are primed by tumor or virus antigens presented 107 by dendritic cells”. This level of specificity seems out of place when discussing general cell movement and migration.

2.        The citation of a book on lines 117-120 can be listed in the references for clarity.

3.        In the section titled "T cell migration to cancer in the perspective of integrins and cell adhesion molecules," although integrins and selectins, as well as ICAM-1, vascular cell adhesion molecule 1 (VCAM-1), ICAM-2, are discussed, the section lacks depth in terms of mechanisms and literature coverage. 

Comments on the Quality of English Language

Moderate editing of English 

Round 2

Reviewer 2 Report

Comments and Suggestions for Authors

In this substantial revision, the authors have adequately addressed all of the concerns I raised during the first round of review. The extensive additions and revisions have significantly broadened the scope of knowledge and literature coverage. The incorporation of three additional figures illustrating the migration of both adaptive and innate immune cells into tumors effectively captures the complexity of these biological mechanisms. This enhanced content is impressive and greatly aids readers striving to gain a deeper insight into this intricate yet captivating phenomenon. In summary, the manuscript has significantly improved in both quality and structure. I have no additional concerns or comments to add.